# Arthropod species loss underpins biomass declines

**Benjamin Wildermuth** [1,2] ✉, **Maximilian Bröcher** [1], **Emma Ladouceur** [2,3,4,5], **Sebastian T. Meyer** [6], **Holger Schielzeth** [1,2], **Michael Staab** [7,8], **Rafael Achury** [6], **Nico Blüthgen** [7], **Lionel Hertzog** [9], **Jes Hines** [2,10], **Christiane Roscher** [2,11], **Oliver Schweiger** [2,12], **Wolfgang W. Weisser** [6] & **Anne Ebeling** [1,2]

Recent declines in arthropod diversity, abundance and biomass are central to the global biodiversity crisis. Yet, we lack a mechanistic understanding of the respective contributions of species richness, species identity and abundance to overall biomass change, and how the environment filters these processes. Synthesizing 11 years of data from a biodiversity experiment and from farmed grasslands in central Europe across a gradient of plant species richness and land-use intensity, we show that local arthropod biomass declines were predominantly (>90%) linked to species richness losses. Abundance declines among persisting species accounted for only 5–8% of lost biomass. The role of species identity depended on the environment and diminished over time: especially under high plant diversity and low land-use intensity, arthropod species with both below-average total biomass and above-average individual biomass (large, rare species) contributed disproportionately to species turnover—but this was only detectable in early years when the communities were still relatively abundant. We conclude that arthropod communities are currently homogenizing towards few common species of similar biomass, probably reducing their adaptability to future environmental change. Increasing the diversity and reducing the land-use intensity of grasslands may mitigate ongoing community simplification and loss of arthropod diversity and functioning.

Amid the global biodiversity crisis and the related loss of ecosystem functioning, arthropods are receiving increased attention[1–4]. Arthropods are the most diverse and abundant animal group on Earth[5], but their numbers are decreasing at concerning rates[6–10], which may escalate further with ongoing climate change[11]. While the causes of arthropod declines are often related to anthropogenic global change, including land-use intensification and subsequent loss of habitat and basal resource diversity[2,4,12,13], the consequences of their shrinking populations on ecosystem functioning are poorly understood[14]. Numerous arthropod-mediated ecosystem functions, such as energy flow between trophic levels, are strongly influenced by

their biomass[15,16]. With an estimated total biomass similar to that of humans and their livestock[17], terrestrial arthropods play pivotal roles in food webs and nutrient cycling[16]. Reported declines of arthropod biomass[2,8,18–21] may therefore impede ecosystem functioning and stability across trophic levels[2,15].

Notably, recent studies have also reported neutral or positive site-level temporal trends of arthropod species richness, abundance and biomass[22–24]. However, even if there are no local declines in species richness, abundance or biomass, community (dis-)assembly must be taken into account to capture changes in species identities and dominance, potentially altering ecosystem functions provided by

arthropods[25,26]. This is for multiple reasons: (1) homogenization: local species richness measures may miss homogenizing effects on the functions present in the community, for example, due to adaptation to specific land-use types or novel climatic regimes[23,27]; (2) trait shifts: trait-based analyses suggest that anthropogenic global change may increase shares of small-bodied species in arthropod communities, possibly because species with smaller body sizes are better able to cope with diminishing, yet variable, resource and habitat availability[28,29]; and (3) abundance shifts: the ecological consequences of community turnover moreover depend on abundance changes in persisting species, that is, shifts in dominance[30], and the abundance of lost and gained species[26,31,32]. For example, formerly highly abundant species may not be lost entirely, but declining numbers could reduce their functional impact substantially[26]. On the other hand, rare species with small contributions to the communities' functioning may be lost entirely, but the consequences for the net community functioning could be negligible[25,26]. Indeed, rare species are generally at higher risk of declining than common or dominant species[6,33] (but see refs. 10,21,34), potentially shifting the relationship between species richness and ecosystem functioning over time[35]. In sum, combining the quantitative and qualitative perspective of abundance change and species identity turnover within community assembly may help elucidate shifts in community metrics, such as biomass, and potentially associated ecosystem functioning, that previously went unnoticed[20,26,30,36].

Other than the intrinsic community (dis-)assembly processes of declining arthropod numbers, accelerating and mitigating factors of the decline need to be identified[13]. Widespread negative effects of climate change on biodiversity are well documented[1–3,11,37], but local plant diversity declines and land-use practices also affect ecosystems[13,38]. On average, diverse plant communities benefit arthropod communities, increasing their (multitrophic) diversity[38,39], stability[40] and functioning[41,42]. Plant species diversity may be especially important for primary consumers such as herbivores, but indirect effects can also escalate up to higher trophic levels such as predators[39,43]. Land-use intensification, including, for example, fertilization, frequent mowing and more intense grazing in grasslands, however, can homogenize arthropod communities[27,32], accelerating the global loss of species and ecosystem functioning[13,14]. It is therefore crucial to examine the role of plant diversity and land-use intensity (LUI) in shaping temporal arthropod community (dis-)assembly and functioning.

Here we used two time series of highly standardized arthropod samplings over periods of 11 years each. One time series (2010–2020; Coleoptera, Hemiptera, Araneae, Hymenoptera) was collected from the Jena Experiment, an experimental grassland site in central Germany, comprising 80 small-scale plots (5 × 6 m) along a controlled gradient of plant species richness (PSR)[44]. The other time series (2008–2018; Coleoptera, Hemiptera, Araneae, Orthoptera) comes from 150 grassland plots of larger size (50 × 50 m) in the Biodiversity Exploratories, a network of real-world farmed grasslands spanning a wide range of management practices in three geographic regions across Germany[45]. Separately for each time series, we used the ecological Price equation to partition temporal changes in local arthropod community biomass into the contributing components of community dis-(assembly). The Price equation was originally developed for quantifying changing gene frequencies under natural selection[46]. The ecological adaptation partitions changes in ecosystem functions[25,26,47], or in our case biomass[48,49], between two communities into the underlying community (dis-)assembly processes, separating effects of average species turnover (species richness) from non-average species turnover (species identity) and effects independent from species turnover (here: abundance change). Specifically, the five components are: (1) + (2) species losses and gains assuming that all species undergoing turnover have average biomass relative to their respective communities (expected effect of species richness); (3) + (4) the difference between the expected and observed biomass change associated

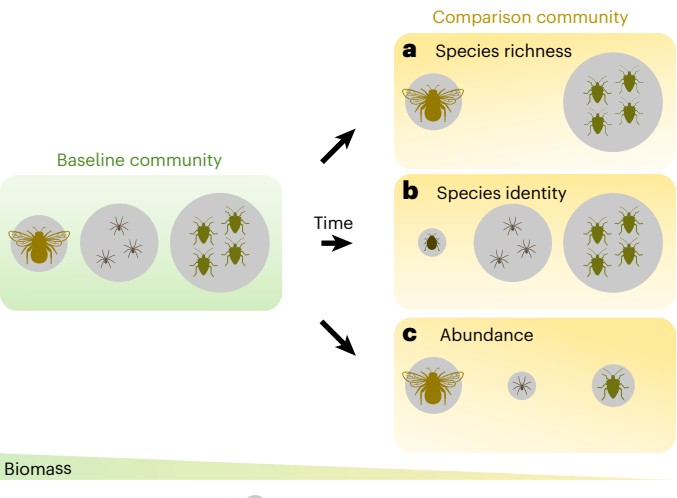

**Fig. 1 | Different scenarios of temporal changes in arthropod community assembly and biomass between a baseline community and a comparison community. a–c**, Changes in the total community biomass can be associated with changes in species richness (**a**), species identity (**b**) and abundance of persisting species (**c**). The species richness component assumes an equal (average) contribution of all species to community biomass. However, species under turnover may have non-average biomass (**b**), in which case their identity must be considered. Consequently, the species identity component reflects the difference between the biomass change expected from the species richness component and the actual observed biomass change associated with species turnover. In comparison community **a**, one species with average total biomass relative to the baseline community was lost (the expected species richness change explains the observed biomass change); in comparison community **b**, one species with below-average total biomass was lost and one species with even lower biomass was gained (species identity change has to be considered to explain the observed biomass change); and in community **c**, no species was lost or gained, but the abundances of two species declined (abundance changes of persisting species explain the biomass change). All scenarios can occur in combination (see Extended Data Fig. 1 for an illustrated example). Reversing the roles of baseline and comparison in this hypothetical scenario would illustrate opposite trends of community and biomass changes. Credit: arthropod icons, Gabriele Rada/iDiv.

with species turnover, that is, the deviation of lost and gained species from the average biomass of their respective communities (species identity of lost and gained species); and (5) changes in abundance of persisting species[25] (Fig. 1 and Extended Data Figs. 1 and 2; see Supplementary Methods for the mathematical equation). In declining arthropod communities, the ecological Price equation may thus help to identify whether species loss per se or more subtle changes in community composition underpin biomass loss. Because high inter-annual variability of arthropod diversity, ecosystem functioning and environmental conditions was previously reported in both research programmes[43,50], we modelled linear temporal trends of each component based on pairwise comparisons using a restricted moving average approach. For this, we pooled all available pairs for each time span (moving average) that include any of the first 5 years as baseline (restriction). We thus generated more generalizable results, reducing the sensitivity to the first sampling year and single years in general (with, for example, climatic extremes or random events affecting sample size) while also reflecting systematic temporal trends[51]. Given the expected variability between any two community-level samples due to detection probabilities[52,53], we included a control in our analysis, quantifying intra-annual species turnover and associated biomass change among replicates[25]. In interaction with time, we also assessed the effects of PSR (Jena Experiment) and LUI (Biodiversity Exploratories; based on mowing, grazing and fertilization[54]) on

arthropod responses. Assuming a more pronounced decline of secondary consumers, such as predators, due to bottom-up effects[7,21,43], but a more direct link of plant diversity to primary consumers, such as herbivores[27,55], we further added separate analyses for the community (dis-)assembly of herbivorous and predatory arthropods, using the Jena Experiment data.

## Results and discussion
### General patterns of biomass decline
Analysing a total of 239,690 arthropod individuals across 1,572 morphospecies (Supplementary Tables 2 and 3), we found an overall decline of local arthropod biomass and species richness in central European grasslands over time in both time series (Jena Experiment: ~5% yearly declines; Biodiversity Exploratories: ~0.5% yearly declines; but see refs. 8,43). In the beginning of both time series and among intra-annual control comparisons, species gains compensated for species losses, but gain rates stagnated or decreased over time, while species losses increased (Figs. 2a,b and 3a,b, and Extended Data Fig. 3). After 7 years, more than 90% of predicted local arthropod biomass loss was associated with species richness declines, while abundance losses of persisting species contributed only up to 8%. Species identity mattered most in early years: detected species with below-average total biomass and above-average individual biomass (that is, mostly rare species) contributed disproportionately to species turnover (Figs. 2c,d and 3c,d). In later years, however, lost and gained detected species had more average biomass, indicating that rare species were lost early on or decreased in abundances to the point of undetectability, while the community structure simplified towards few common species—which themselves were increasingly lost. Increasing PSR and decreasing LUI generally promoted absolute arthropod community turnover (Figs. 2 and 3), mitigating effects of community simplification.

Estimated yearly biomass declines of arthropods were approximately 5.1% (95% confidence intervals (CI): 2.7, 7.4) in the Jena Experiment and 0.5% (0.1, 1.1) in the Biodiversity Exploratories (Figs. 2f and 3f, and Supplementary Tables 4–8). Species richness declines were ~4.7% (2.7, 6.6) and ~0.2% (0.03, 0.4), respectively (Extended Data Fig. 3 and Supplementary Tables 4–6). These rates are lower than previously reported annual decline rates of approximately 7% for biomass and 3.8–5.6% for species richness in both research programmes[8,43]. Our use of a restricted moving average approach was expected to yield lower estimates than time series with fixed baselines due to averaging of systematic temporal trends within each moving window. Yet, reducing the dependence on the first sampling year (baseline) and smoothing, but not removing, extreme years with for example climatic anomalies makes detected trends more generalizable and robust[51] (see Extended Data Figs. 4 and 5, Supplementary Notes 1 and 2, and Supplementary Figs. 1 and 2 for further sensitivity analyses). Higher decline rates in the Jena Experiment than in the Biodiversity Exploratories may partly be explained by the highly controlled setting with a constant management regime in the Jena Experiment, and greater temporal variability of management intensity and arthropod biomass in the Biodiversity Exploratories. Notably, moderate plant biomass declines due to nutrient depletion and hence lower resource availability in the Jena Experiment[43,56] did not show a significant relationship with arthropod biomass loss (Supplementary Table 9). Extinction debts owing to the establishment of the Experiment on a previously agricultural site in 2002 are also unlikely to have substantially influenced our results, as they were probably largely paid off by 2010, when grassland arthropod communities were established[57]. However, our partitioning analysis focuses on relative arthropod community (dis-)assembly processes driving biomass change rather than providing absolute numbers of the total decline. Overall, our study adds moderate, but robust, support to the growing evidence of declining arthropod diversity and biomass[2,6,18,20,34]. Results from alternative modelling analyses, that is, fixed baseline comparisons and unrestricted moving average comparisons (Methods), were similar to the reported main analysis (Extended Data Figs. 4 and 5, and Supplementary Note 1).

### Community assembly and biomass decline
Despite the differences in setup (experiment versus real world), spatial scale, location and taxonomic coverage in the investigated research programmes, our partitioning approach—that is, the ecological Price equation—consistently showed that the vast majority of local arthropod biomass loss was linked to declines in species richness (Fig. 4). Even when accounting for species identity effects, 95.3 (Jena Experiment; 95% CI: 72, 116.4) and 93.5 (Biodiversity Exploratories; CI: 43.1, 150.3) of the total biomass loss after 7 years was associated with species loss (Supplementary Tables 7 and 8). Abundance declines in persisting species were associated with relatively small biomass losses of 4.8% (1.6, 10.5) and 8.1% (2.5, 19.1). Species identity effects, defined as the deviation in biomass of lost and gained species from the expected biomass change based on species richness (assuming all species have average biomass relative to their respective communities), played a dynamic role over time: initially, species with below-average total biomass but above-average individual biomass contributed disproportionately to species turnover (Figs. 2c,d and 3c,d, Extended Data Figs. 6 and 7, and Supplementary Tables 5–8, 10 and 11). These species can be considered rare, because the low total biomass contribution despite the high individual biomass is a consequence of their low abundances. Indeed, large-bodied species are typically rare[58] (Supplementary Note 3). The initial disproportionate turnover of rare species offset the expected biomass change of species richness losses by up to 31.6% (95% CI: −16.7, 82; Figs. 2c,d and 3c,d, and Supplementary Tables 7 and 8). Yet, while species loss and associated biomass loss increased in later years, the absolute offsets by lost rare species stagnated or declined, reducing relative offsets to 0.4 (−0.1, 2.8)–3.4% (0.4, 9), and gained species had almost entirely average biomass (see Supplementary Fig. 3 for additional rank abundance curves). Underscoring the diminishing role of species identity and, particularly, of (relatively) rare species over our study period, biomass change associated with detected spatial turnover of rare species (below-average total biomass and above-average individual biomass) among replicates declined significantly over time (Extended Data Figs. 8 and 9, and Supplementary Tables 12–15). This is notable because rare species are statistically more likely to show turnover among communities than common species[33,35]. We suggest that, because rare arthropod species in grasslands were previously shown to decline the fastest[8], their diminishing contribution to species turnover and biomass change is a symptom of their decline. Note, however, that supplementary analyses showed that our sampling coverage slightly decreased over time from 89 to 85% (95% CI: ±0.5%) in the Jena Experiment and 90 to 89% (±0.4%) in the Biodiversity Exploratories, potentially reducing the detection of rare species (Supplementary Fig. 4, and Supplementary Tables 16 and 17). Yet, we emphasize that first, because our sampling effort was constant over time, declining detections still reflect shrinking abundances and biomass, indicating that most species are becoming rarer in absolute terms; and second, declines in the turnover of rare species were particularly strong in the Biodiversity Exploratories, where sampling coverage was higher than in the Jena Experiment and only decreased by 1% (Fig. 3c,d and Supplementary Fig. 4). Further supplementary analyses showed that rarely detected species were not just highly mobile 'tourists' in our plots, but rather the opposite: commonly detected species tended to show higher mobility (Supplementary Fig. 5 and Supplementary Tables 18 and 19). This implies that in the open systems of our plots, turnover of common species may be amplified by mobile, visiting species to some degree (see Supplementary Note 4 and Supplementary Figs. 6 and 7 for additional sensitivity analyses on the robustness of observed patterns when reducing the analysis to species that occurred in at least 10% or 30% of all plots per year per research programme). We thus add robust findings of temporally homogenizing

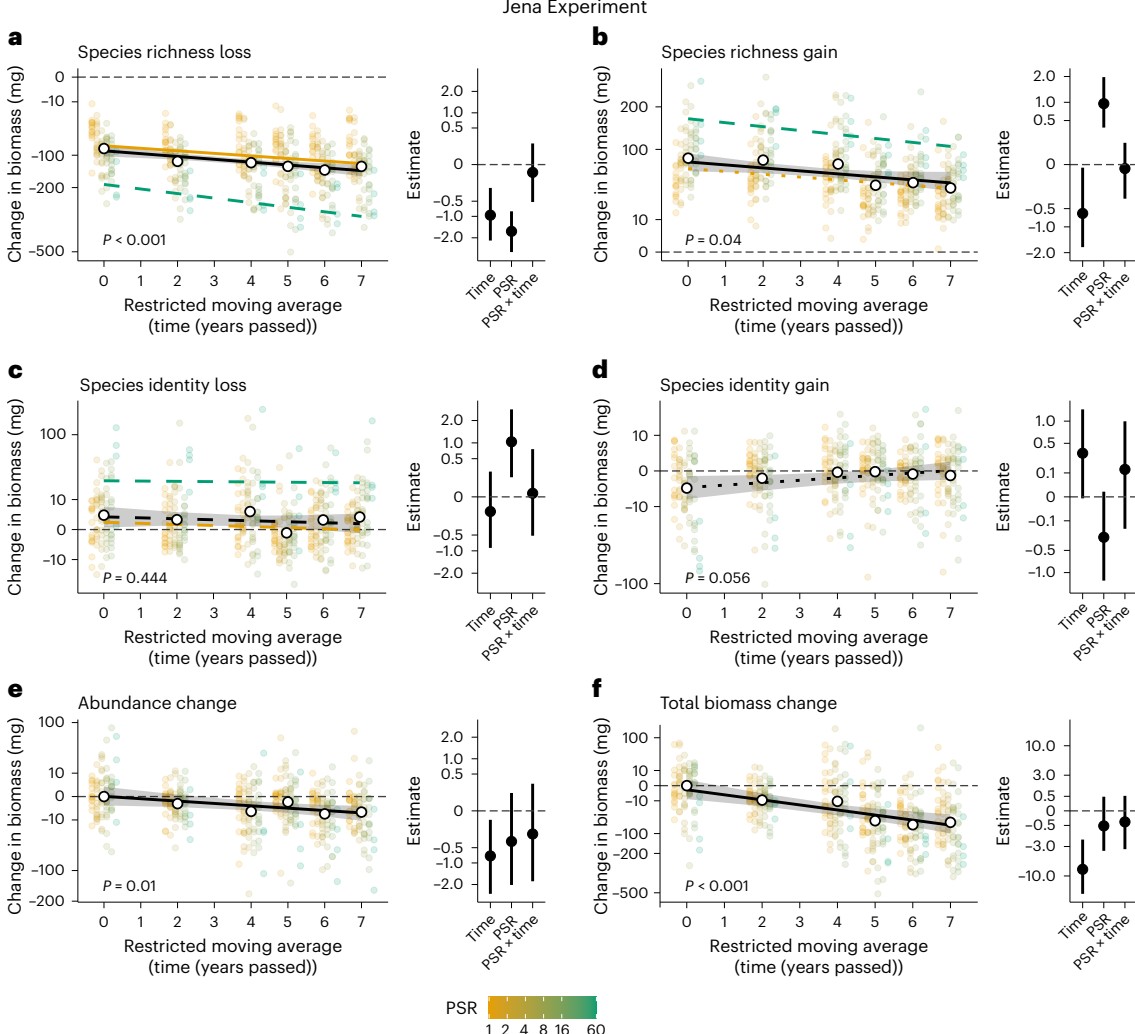

**Fig. 2 | Arthropod biomass declines are primarily associated with species richness loss, while species richness gains and the role of species identity decline over time in the Jena Experiment.** All panels are based on replicate-level ($n = 160$) median point estimates and predictions from 1,000 linear mixed-effects models drawing from shuffled data subsets, avoiding the reuse of sampling events in multiple pairwise comparisons. **a**–**e**, Main plots show temporal biomass change per replicate per plot (restricted moving average) associated with species richness loss (assuming average biomass of lost species relative to their respective communities; **a**), species richness gain (assuming average biomass of gained species relative to their respective communities; **b**), species identity loss (deviation of observed biomass change associated with lost species from the expected biomass change associated with species richness loss; **c**), species identity gain (deviation of observed biomass change associated with gained species from the

expected biomass change from species richness gain; **d**) and abundance change of persisting species (**e**). **f**, Total biomass change without partitioning. Year 0 represents biomass change within years, between replicates (control). Mean biomass change values per replicate per plot are shown as coloured dots along a plant species richness (PSR) gradient (legend), white dots show mean values across all plots ($n = 80$). Solid regression lines indicate significant relationships ($P < 0.05$), dotted lines indicate marginally significant relationships ($P < 0.1$), dashed lines indicate non-significant relationships ($P \geq 0.1$); exact two-sided $P$ values for the main effect of time are provided in the panels. Shaded areas around the black main effect line represent 95% CI. Significant effects of PSR on arthropod biomass change are shown in the main panels, coloured along the PSR gradient (legend). Insets show median point estimates with 95% CI (error bars) for the effects of time and PSR on biomass change individually and in interaction (×).

biomass distributions in arthropod communities to previous reports on homogenizing taxonomic and functional diversity[23,59]. Our analyses suggest that this is both driven by homogenizing abundance distributions—that is, declines of rare species[8,32]—and homogenizing size distributions—that is, declines of large-bodied species[21,28]. Losses of rare species with potentially unique and complementary functional profiles threaten ecosystem functioning and resilience[31,35]. At the same time, the overwhelming contribution of species richness loss per se to declining arthropod biomass reveals the consequences of arthropod communities increasingly losing their common members[10,21]. This may have further escalating negative implications for multitrophic diversity and ecosystem functioning[7,10,15].

In our datasets, species gains were increasingly unable to compensate for species losses (Figs. 2b, 3b and 4, and Extended Data

Fig. 10). Notably, species losses and gains in moderate sample sizes, such as ours, always depend on detection probabilities rather than real extinctions or immigrations[52], and a key limitation of the ecological Price equation is that information on species-level traits and identities are lost in the community aggregation[25]. Nonetheless, increasing imbalance in detected species gains and losses indicates real community-level species loss and counters previous observations of balanced extinctions and immigrations in changing arthropod communities[4,59]. We underscore the lack of species gains because immigrations sometimes outpace extinctions in community (dis-)assembly under a fast-changing climate[60]. If gains do not keep up with losses already now, species declines are likely to continue. Overall, the partitioning shows that local arthropod biomass declines in grasslands are strongly associated with species richness declines, supporting the

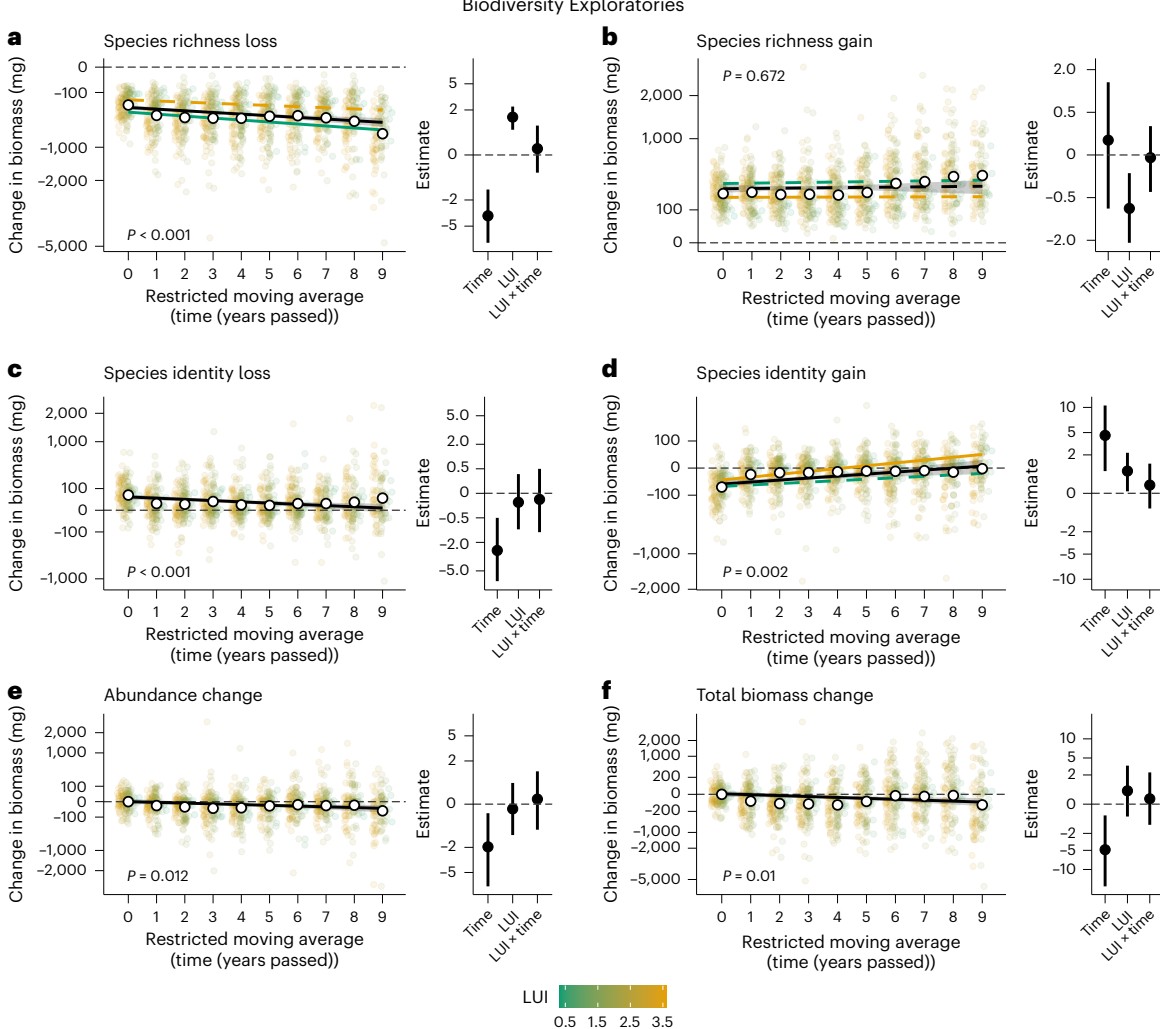

**Fig. 3 | Arthropod biomass declines are primarily associated with species richness loss, while species richness gains stagnate and the role of species identity declines over time in the Biodiversity Exploratories.** All panels are based on replicate-level (*n* = 300) median point estimates and predictions from 1,000 linear mixed-effects models drawing from shuffled data subsets, avoiding the reuse of sampling events in multiple pairwise comparisons. **a**–**e**, Main plots show temporal biomass change per replicate per plot (restricted moving average) associated with species richness loss (**a**), species richness gain (**b**), species identity loss (**c**), species identity gain (**d**) and abundance change of persisting species (**e**; see Fig. 2 for detailed explanations). **f**, Total biomass change without partitioning. Year 0 represents biomass change within years, between

replicates (control). Mean temporal biomass change values per replicate per plot are shown as coloured dots along the land-use intensity (LUI) gradient (legend), white dots show mean values across all plots (*n* = 150). Solid regression lines indicate significant relationships (*P* < 0.05), dashed lines indicate non-significant relationships (*P* ≥ 0.1); exact two-sided *P* values for the main effect of time are provided in the panels. Shaded areas around the black main effect line represent 95% CI. Significant effects of LUI on arthropod biomass change are shown in the main panels, coloured along a LUI gradient (legend). Insets show median point estimates with 95% CI (error bars) for the effects of time and LUI on biomass change individually and in interaction (×).

validity of biodiversity–ecosystem functioning relationships based on species richness[42,58]. We emphasize, however, that our results reflect on local-scale community turnover (alpha diversity). With increasing spatial scale, temporal species turnover statistically decreases and shifts in community composition (species identity) and particularly abundances, that is, changes in species dominance, may dominate community turnover[30]. In our study, this is reflected by slightly higher contributions of species identity and abundance to biomass change in the 50 × 50 m plots of the Biodiversity Exploratories (3.4%, 8.1%) than in the 5 × 6 m plots of the Jena Experiment (0.4%, 4.8%). Moreover, supplementary analyses show that removing species occurring in less than 10% or 30% of plots reduced biomass change associated with species richness turnover, but the relative contribution of abundance change increased in both research programmes by 1–2%, and species identity effects increased by ~1% in the small plots of the Jena Experiment (Supplementary Note 4, Supplementary Figs. 6 and 7,

and Supplementary Tables 20 and 21). Removing the rarest detected species can thus be interpreted as similar to increasing the sample size: while stochastic species turnover decreases, shifts in abundance and species identity gain in importance[30]. Yet, removing the rarest detected species decreased the absolute and relative contributions of rare species turnover in the Biodiversity Exploratories. This emphasizes differences in the study designs, with the Biodiversity Exploratories spread over a large geographic range and the Jena Experiment covering one field site; while 72% of species occurred in at least 30% of the plots in the Jena Experiment, only 57% of species did so in the Biodiversity Exploratories. Accordingly, excluding species that occur in few plots was more impactful in the Biodiversity Exploratories. In sum, species richness alone cannot fully explain changes in arthropod biomass across spatial scales[20,36]. Species identity of lost species may counteract biomass declines as rare species tend to be lost first, especially in diverse communities[6,33].

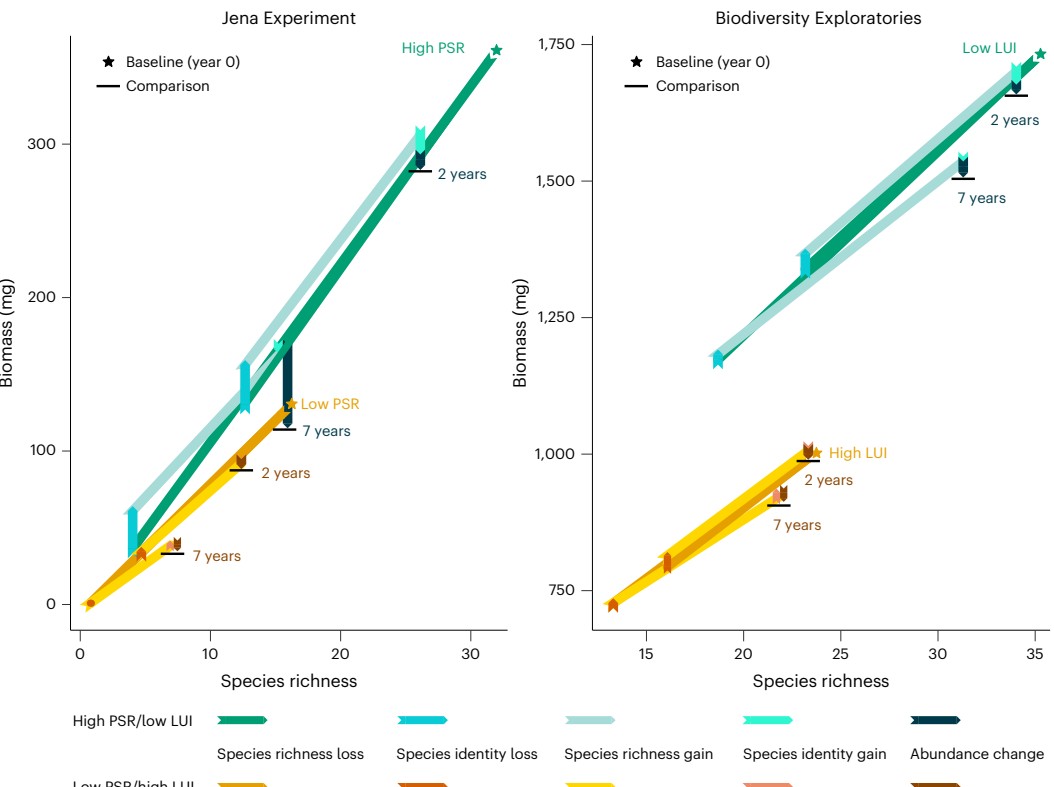

**Fig. 4 | Arthropod biomass declines are primarily associated with species richness loss, while losses and gains of species with non-average total biomass (species identity) contribute disproportionately to arthropod biomass change under high PSR (Jena Experiment) and low LUI (Biodiversity Exploratories).** Modelled change of arthropod biomass and species richness per replicate per plot after 2 and 7 years (restricted moving average) in dependence on plant species (PSR) and land-use intensity (LUI). Starting at the average community biomass and species richness value in the first year (baseline), the community assembly components of biomass change are displayed as vectors (arrows) in the order of (1) species richness loss, (2) species identity loss, (3) species richness gain, (4) species identity gain and (5) abundance change of persisting species (see Fig. 2 for detailed explanations), reaching the predicted

comparison community values of absolute biomass and species richness after the respective time spans of 2 and 7 years. Most arthropod biomass change is associated with species richness change, but the vertical vectors of species identity and abundance change show that species richness change alone does not explain all biomass change. Vectors are based on median predictions from 1,000 linear mixed-effects models (see Methods and Supplementary Tables 22 and 23). Vectors for high PSR (60 plant species) and low LUI (0.5) plots are coloured in green/blue, vectors for low PSR (monoculture) and high LUI (3.5) plots are coloured in beige/brown (see legend). See Extended Data Fig. 10 for an illustration of the underlying data spread after 7 years. Seven years is the maximum replicated moving window in the Jena Experiment, see Supplementary Note 1 for information on the maximum time span of 10 years (fixed baseline).

## Environmental drivers

We found that arthropod biomass loss and the roles of species richness and species identity were mediated by plant diversity and LUI. Arthropod communities in plots of high PSR (Jena Experiment) and communities in plots of low LUI (Biodiversity Exploratories) showed higher absolute species and biomass turnover with disproportionate contributions of rare species (Figs. 2–4, and Supplementary Tables 5, 6, 22 and 23), suggesting larger species pools with more heterogeneous biomass and abundance distributions[55]. Especially in the small plots of the Jena Experiment, we cannot assume that the sampled arthropods completed their full lifecycle in high-diversity plots, but we can identify a stark and consistent spatial preference, indicating enhanced provision of resources and habitat[41,55] (Figs. 3d and 4). Meanwhile, communities under low PSR and high LUI contained simplified communities, with their biomass largely concentrated in common species. In plots of high LUI, gained species even had above-average total biomass, possibly indicating the rise of abundant generalist species, establishing their numerical dominance[27] (Figs. 3d and 4). This provides temporal support for findings along space-for-time gradients, reporting taxonomic and functional homogenization of arthropod communities with decreasing plant diversity and increasing LUI[27,38,55]. A possible avenue for mitigating the ongoing community simplification is, therefore, a diversification of plant communities via, for example, reduced LUI or active restoration

in managed grasslands[41,43,61–63]. Yet, absolute (not relative) biomass declines were especially pronounced in the diverse arthropod communities associated with high PSR and partly also under low LUI (Figs. 2a,e,f and 3a). This indicates that landscape-scale environmental conditions may impose such strong negative effects that locally beneficial conditions can only buffer biodiversity and functioning declines to a limited extent[3,8]. Accordingly, arthropod declines were previously not only reported from intensively managed or disturbed ecosystems, but also from protected and natural grasslands[18] or tropical rainforests[2,37].

## Patterns across trophic guilds

Temporal trends or responses to plant diversity of primary consumers (herbivores) and secondary consumers (predators) were similar (Fig. 5 and Supplementary Table 24). This adds to the evidence that different trophic guilds are jointly declining[8,12,43]. In trend, plant diversity strengthened the role of species identity for biomass changes in both trophic guilds, but it only promoted rare species turnover significantly for predators (Fig. 5c,d). This contrasts with previous findings that primary consumers are more tightly linked to plant diversity and LUI than secondary consumers[27,55]. Nevertheless, recent research reported that top-down control by arthropod predators increases with plant diversity[64] and declining LUI[65]. Predators may benefit two-fold from increased plant diversity, as both resource diversity and

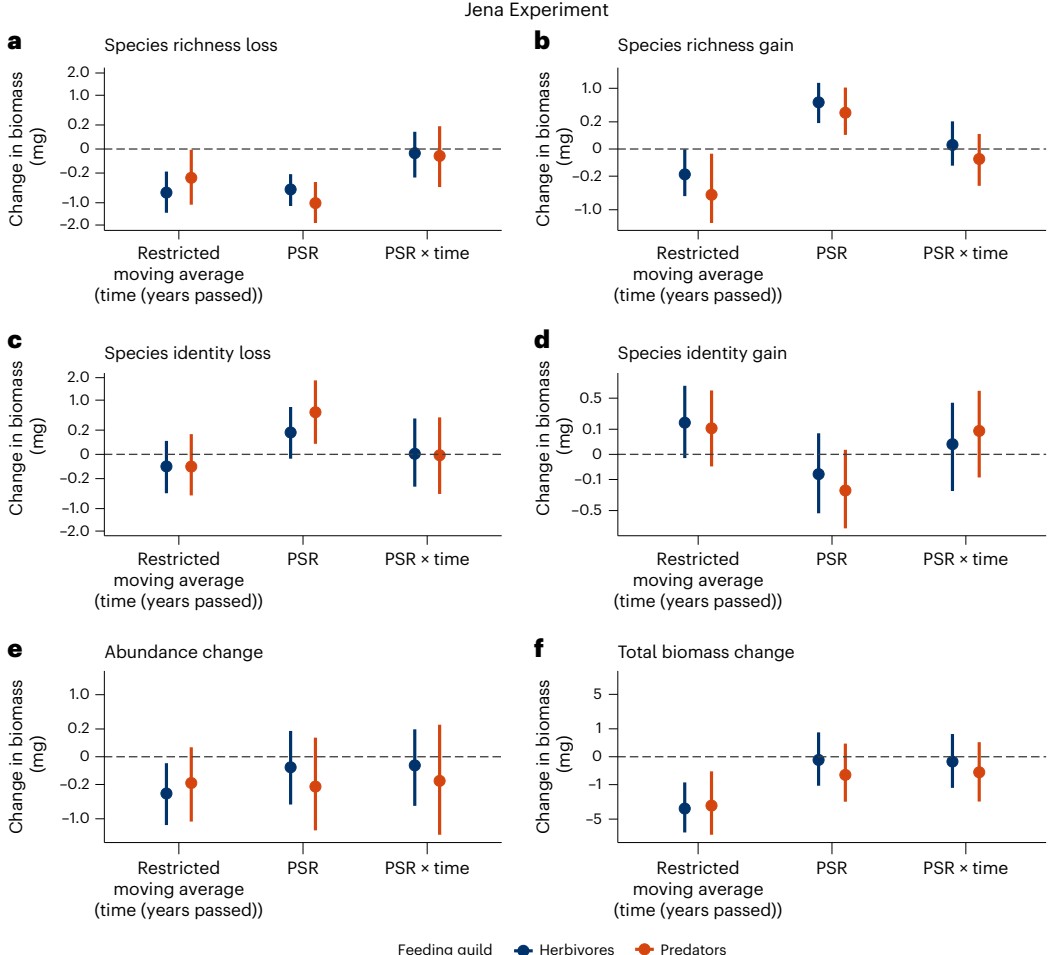

**Fig. 5 | Temporal trends and PSR effects on arthropod biomass change are similar among trophic guilds (herbivores, predators) in the Jena Experiment.** All panels show replicate-level ($n$ = 160) median point estimates from 1,000 linear mixed-effects models drawing from shuffled data subsets, avoiding the reuse of sampling events in multiple pairwise comparisons with 95% CI (error bars). **a**–**e**, Temporal biomass change (restricted moving average) is associated with species richness loss (**a**), species richness gain (**b**), species identity loss (**c**), species identity gain (**d**) and abundance change of persisting species (**e**; see Fig. 2 for detailed explanations). **f**, Total biomass change without partitioning. The point estimates of plant species richness (PSR) effects on biomass change are shown individually and in interaction (×) with time. Estimates for herbivores are coloured dark blue, predators are coloured orange (see legend).

habitat heterogeneity increase—boosting their chances for successful foraging and reproduction[39,66]. Biomass losses associated with abundance declines of persisting species accelerated over time only for herbivores, possibly reflecting their larger sample size[30,55]. Overall, we show that high PSR benefits multitrophic diversity and potentially biomass-mediated functioning[27,39,62].

## Conclusions

Our study provides insights into temporal declines of local arthropod biomass in anthropogenic central European grasslands within a limited and relatively recent time frame (2008–2020). Within that time, we detected a small, yet notable, role of species identity in early years, with rare species disproportionately contributing to species turnover. However, communities shrank and their biomass distribution homogenized almost completely in later years, with biomass declines primarily associated with species richness declines—a pattern that was consistent among the different setups and scales of the Jena Experiment (experiment, small scale) and the Biodiversity Exploratories (real world, larger scale), and which probably transfers to many shrinking biotic communities[30]. These are concerning findings, hinting at simplified biomass distributions and highly vulnerable arthropod communities in the face of ongoing anthropogenic global change, with arthropod species richness per se requiring high priority in conservation efforts.

Yet, in other ecosystems around the world, the mechanisms behind arthropod biomass declines may be different and similar studies are needed elsewhere. Notably, previous long-term research across agricultural, grassland and forest sites suggests that arthropod declines started well before our study period[2,12,19], potentially explaining the already relatively minor role of rare species in the early years of our study. However, long-term studies also indicate a deceleration of the decline in the twenty-first century[3,12]. Our results on fluctuating arthropod biomass in the Biodiversity Exploratories partly support these findings, but the general species turnover and biomass trends emphasize ongoing arthropod community simplification[8,23,59], and that even common species may be increasingly under threat[10,21,34]. Escalating negative consequences for ecosystem functioning have to be expected, as biomass-mediated functions such as energy transfer between trophic levels may be impaired[7,15]. Considering that community simplification can partly be irreversible due to newly established dominance structures[67], swift action is required to halt ongoing arthropod declines[1]. As our results reflect local, community-scale processes, we encourage future research to apply the ecological Price equation to temporal arthropod biomass and diversity change at larger spatial scales, potentially yielding different assembly mechanisms for, for example, whole regions or biomes[30]. On a positive note, we clearly identify increasing plant diversity and decreasing LUI as mitigating factors

for arthropod declines and their community simplification. As plant diversity and decreasing LUI of grasslands are closely intertwined[61], land-use extensification—but not abandonment[68]—is a promising avenue for fostering arthropod-mediated ecosystem functioning and resilience in the face of future environmental change.

## Methods

### Data sampling

The study builds on data collected in grassland plots of the Jena Experiment[44] and the Biodiversity Exploratories[45] in Germany. The Jena Experiment was established in 2002 and consists of one 1 ha field site (central Germany, 130 m above sea level (asl)) with 80 spatially randomized plots of 5 × 6 m each (minimum distance between plots 4 m), comprising unique plant compositions with experimentally implemented PSR levels of 1, 2, 4, 8, 16 and 60 plant species per plot[44]. Monocultures and 16-species mixtures are represented on 14 plots each, 60-species mixtures on 4 plots and all other combinations (2, 4 and 8 species) on 16 plots, while the randomized design prevents clustering of plots of the same diversity. Plant species not included in the originally sown plant composition of a plot are removed two to three times per year. Established in 2006, the Biodiversity Exploratories are observational research plots distributed over three regions (northeast Germany, 3–140 m asl; central Germany, 285–550 m asl; and southwest Germany, 460–860 m asl), with 50 grassland plots in each region (150 plots total, 50 × 50 m each, minimum distance between plots 200 m; see Supplementary Figs. 8 and 9 for maps of both study programmes). The various types and degrees of land use across the observatories range from low to intensively managed grasslands[45]. LUI values were calculated based on the combined intensities of fertilization, grazing and mowing, with LUI values ranging between 0.5 (for example, less than one mowing event per year and no fertilization or low livestock densities) and ~3.5 (for example, frequent mowing and high fertilization or intense grazing)[54]. Because local farmers may change land-use practices among years, we averaged the LUI for the sampling year and the previous 2 years for each sampling[69].

Plots of the Jena Experiment were sampled annually from 2010 to 2020, except for the years 2011, 2013, 2015 and 2018. Arthropods were collected in spring (May) and summer (July) using suction sampling on a volume of ~0.75 m³ at two distinct locations per plot (replicates A and B; minimum distance 1 m; Supplementary Fig. 8). In May 2019, only half of the sampling could be completed; we therefore extrapolated the sample using resampling (see below). In the Biodiversity Exploratories, arthropods were sampled in spring (June) and summer (August) from 2008 to 2018 by sweep netting along 150 m transects with 60 double sweeps per sampling (Supplementary Fig. 9). All samplings were conducted under standardized conditions (dry and windless, after morning dew had dried; see Supplementary Tables 25 and 26 for details on mean temperature and humidity on the sampling day).

The study design of the Biodiversity Exploratories does not include replicates per sampling campaign, because each plot was sampled once at each timepoint. Partitioning biomass change over time, however, requires a control accounting for variability of the community composition and biomass due to detection probabilities within each timepoint[25,53]. To create replicates for both the Jena Experiment and the Biodiversity Exploratories, we split each plot-level arthropod sample into two subsamples. First, we randomly sampled a Poisson-distributed number of individuals without replacement based on half the sample size (replicate A). The remaining individuals were assigned to replicate B. For the incomplete sampling of May 2019 in the Jena Experiment, we kept the whole sample as replicate A and created an artificial replicate B by randomly resampling a Poisson-distributed number of individuals with replacement from the entire sample. To ensure that the artificial replicates A and B captured the real sampling 'noise' within each treatment, we compared their results with those from the original

field replicates in the Jena Experiment and found no differences (Supplementary Fig. 10).

Among all sampled arthropods, we analysed three common arthropod taxa in both research programmes: the highly diverse order of Coleoptera, the herbivore-dominated Hemiptera (excluding Sternorrhyncha) and the predatory Araneae. Hymenoptera (representing diverse feeding guilds) were analysed exclusively in the Jena Experiment (excluding Formicidae), while Orthoptera (mostly herbivorous) were included only in the Biodiversity Exploratories. Overall, these groups can be adequately captured in grasslands with the respective methods[8,55], covering some of the most diverse arthropod taxa and multiple feeding guilds. All taxa were identified at species level, except for Hymenoptera, which were sorted to morphospecies with identification corresponding at least to the taxonomic family. Body length values and feeding guilds were assigned based on ref. 70 and ref. 71. Individual biomass was estimated based on body length, using the taxon-specific allometric equations of ref. 72 at the highest available taxonomic classification (Araneae were split into web-weaving and hunting spiders based on ref. 73).

### Data analysis

The ecological version of the five-part Price equation partitions changes of a given function, or in our case biomass, between a baseline and a comparison community into the components of (1, 2) species richness losses/gains; (3, 4) species identity losses/gains; and (5) changes in the functional contribution of persisting species[25,74] (Fig. 1 and Extended data Fig. 1). Here, the equation therefore first assumes that (1) lost species have average biomass relative to the overall baseline community (species richness loss); and (2) gained species have average biomass relative to the overall comparison community (species richness gain). The component of (3) species identity loss then calculates the deviation of observed biomass change associated with lost species from the expected biomass change associated with species richness loss; and (4) species identity gain calculates the deviation of observed biomass change associated with gained species from the expected biomass change associated with species richness gain. Biomass changes of (5) persisting species can be exclusively attributed to changes in abundance because we used constant body length values from the literature (see above). While the combination of biomass and abundance captures the main acting mechanisms of biomass change in dynamic communities, its species turnover components (1–4) cannot distinguish between effects of abundance and mean individual biomass of a species[25]. Therefore, we supplemented a second approach to the Price Equation, based solely on mean individual biomass per species, removing effects of abundance (see Results and discussion, Extended Data Figs. 2, 6 and 7, and Supplementary Tables 6 and 7).

All analyses were conducted in R 4.2.2 and newer versions[75]. We calculated all components of community change per replicate per plot across all available sampling years and between replicates per plot in each available sampling year, with sampling months pooled. Specifically, we calculated and visualized the five Price components of biomass change (see above) and additionally total biomass change and changes in species richness (species lost, species gained, total species richness change) using the packages priceTools[25] and ggplot2[76]. Because LUI values between years may differ, we averaged the LUI values within each comparison. To assess possible changes in sampling coverage over time, we estimated the sample coverage of each sample (replicate) using the 'Coverage' function with the 'Best' estimator in the entropart package[77]. To control for the influence of species with low sampling probabilities, we added two sensitivity analyses restricting the inclusion in our analyses to species that occur in at least 10% or 30% of all plots per year per research programme (Supplementary Note 4 and Supplementary Figs. 6 and 7).

How community (dis-)assembly drives biomass change over time depends strongly on the baseline community, which itself relies on

environmental conditions before and at the sampling timepoint[48,78]. Basing the analysis on only one baseline can therefore only be interpreted as relative to a specific set of conditions—which are likely to be highly variable over time[43,50]. To identify robust patterns, we applied three analysis approaches: (1) fixed baseline comparison: we contrasted temporal change only to the first year of each time series (2010 in the Jena Experiment, 2008 in the Biodiversity Exploratories). (2) Moving average comparison: we averaged all available time spans categorically, that is, all 1-year comparisons (2008 versus 2009, 2009 versus 2010, …, 2017 versus 2018), all 2-year comparisons, all 3-year comparisons and so on, and all controls. Only time spans with at least two replicates were considered. This provides more generalizable insights into temporal community (dis-)assembly dynamics, less depending on starting conditions and extreme years[51]. The maximum replicated time spans available were 7 years (Jena Experiment) and 9 years (Biodiversity Exploratories). A downside to the moving average approach, however, is that information on systematic temporal changes in community assembly patterns is partly lost. (3) Restricted moving average comparison: to balance the highly sensitive first approach (contrasting versus one baseline year) and the generalizing moving average (contrasting across all available years), we restricted the moving average to comparisons including any of the first 5 years as baseline. This enables robust estimations of a general timeline for community (dis-)assembly processes while safeguarding sensitivity to systematic changes over time (Supplementary Note 1).

Using linear mixed-effects models in the lme4 package[79] and linear models (base R; only used for fixed baseline comparisons), we fitted models for each replicate-level biomass and species richness component as response. We included the temporal differences including controls (between-replicates turnover, that is, year 0) as fixed baseline comparison, moving average comparison and restricted moving average comparison in interaction with PSR (defined as the initially sown PSR; Jena Experiment) or LUI (Biodiversity Exploratories) as scaled continuous fixed effects[80]. We further added plant biomass change as predictor in a supplementary model of total arthropod biomass change in the Jena Experiment, accounting for possible effects of systematically declining plant biomass[43] (Supplementary Table 9; data derived from refs. 56,81). In additional sensitivity analyses, we excluded all control comparisons from the models, analysing solely changes in Price components among increasing time spans (Supplementary Note 2 and Supplementary Figs. 1 and 2). Because our data structure compares each sample across multiple time spans, using all pairwise comparisons in one model would inflate the degrees of freedom, introducing pseudoreplication[82]. Therefore, we randomly sampled 1,000 subsets of the pairwise comparisons, with the sample size determined by the maximum number of possible random pairs without reusing any plot/year combination. We then ran models for each subset. We included the sampling plot and each contributing sampling year of each temporal comparison (year baseline and year comparison) as random effects in the moving average mixed-effects models because both were sampled multiple times across unique combinations. Because for the fixed baseline models, each plot was only picked once per resampling and the temporal difference was covered by the fixed effect (the baseline is fixed, therefore each temporal comparison consists of unique years), we used a simple linear model without random effects. We obtained point estimates for low and high PSR and LUI values using the 'emtrends' function from the emmeans package[83] and linear predictions via the 'ggemmeans' function from the ggeffects package[84]. We derived median point estimates, predictions and 95% CIs from the distribution of the 1,000 model coefficients and predictions[82,85]. We further calculated two-sided *P* values of point estimates as twice the proportion of point estimates falling on the less frequent side of zero[86]. To estimate the combined contributions of (1) species richness losses and gains, (2) species identity losses and gains, and (3) all species turnover components to overall biomass change, we also derived the median and 95% CI for their combined 1,000 predictions. Moreover, to analyse the temporal development of within-year between-replicate species turnover, we fitted a model exclusively for the controls across all years (see Results and discussion, Extended Data Figs. 8 and 9, and Supplementary Tables 12–15). Although the pairwise comparisons in this model do not cover multiple time spans, pseudoreplication was still an issue as each sample was compared twofold, once as baseline and once as comparison (replicate A versus B, and B versus A). Therefore, this model was also run across 1,000 subsets, sampling each plot/year combination once per iteration (either A versus B or B versus A were sampled) and plot was included as random effect in the model. In supplementary models, we fitted (1) the overall sampling coverage per replicate per plot as response, with sampling year in interaction with PSR/LUI as fixed effects, and (2) the abundance per species per replicate/plot as response and species dispersal ability (obtained from refs. 70,71) as fixed effect. The latter model was fitted as a generalized linear mixed-effects model using the Poisson family with log-link to fit the count data distribution[79]. Both models included two random effects: plot nested within year, accounting for repeated measures within plot–year combinations, and plot alone, accounting for plot-level variability. We checked all models for normal and homoscedastic residual distribution and ensured low multicollinearity between predictors using variance inflation factors (VIF < 2; ref. 87). All responses except the sample coverage were square-root transformed to ensure good model fit[88]. Because relative change in community biomass is a symmetric process, that is, positive and negative values are possible with comparable skew (few extreme values), we used absolute values for the transformation and reconstructed the original sign afterwards[89]. For approximations of percentage biomass loss over time, we divided the predicted biomass loss at the last timepoint by the average biomass within the first 5 years of each time series (restricted moving average).

### Reporting summary

Further information on research design is available in the Nature Portfolio Reporting Summary linked to this article.

## Data availability

This work is based on data from the Jena Experiment (DFG research units FOR 5000 and FOR 1451) and the Biodiversity Exploratories (DFG Priority Program 1374). The analysed data from the Jena Experiment are available at https://jexis.idiv.de/ under the identifiers 747 (raw data, available upon request due to ongoing analyses) and 749 (processed data, openly available). The analysed data from the Biodiversity Exploratories are openly available at https://www.bexis.uni-jena.de under the identifiers 32231 (raw data) and 32232 (processed data).

## Code availability

The R code used for the analyses is provided at https://github.com/BWildermuth/Model_every_species_counts.

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

## Acknowledgements

We thank the technical staff of the Jena Experiment for their work in maintaining the experimental field site and many student helpers for weeding of the experimental plots and support during measurements. Further, we thank the speaker of the Jena Experiment, N. Eisenhauer, and the data management team led by Y. Huang. We thank the managers of the three Exploratories, K. Wells, S. Renner, K. Reichel-Jung, I. Steitz, S. Weithmann, S. Gockel, K. Wiesner, K. Lorenzen, J. Vogt, A. Hemp, M. Gorke, M. Teuscher and all former managers for their work in maintaining the plot and project infrastructure; S. Pfeiffer, M. Gleisberg, C. Fischer and J. Mangels for giving support through the central office; J. Nieschulze, A. Ostrowski and M. Owonibi for managing the central data base; and M. Fischer, E. Linsenmair, D. Hessenmöller, D. Prati, I. Schöning, F. Buscot, E.-D. Schulze and the late E. Kalko for their role in setting up the Biodiversity Exploratories project. We also thank E.-D. Schulze and B. Schmid for their role in initiating and establishing the Jena Experiment. We thank the administration of the Hainich National Park, the UNESCO Biosphere Reserve Swabian Alb and the UNESCO Biosphere Reserve Schorfheide-Chorin as well as all land owners for the excellent collaboration. Further, we thank R. Achtziger, E. Anton, T. Blick, B. Büche, F. Creutzburg, M.-A. Fritze, L. Funke, M. Gossner, R. Heckmann, A. Kästner, F. Köhler, G. Köhler, T. Kölkebeck, C. Morkel,

C. Muster, F. Schmolke, T. Wagner and O. Wiche for their tremendous work with identifying arthropod individuals. We thank G. Rada and the German Centre for Integrative Biodiversity Research (iDiv) for supporting the illustrations of the paper. The work has been funded by the German Research Foundation (DFG) via the 'Jena Experiment' research units FOR 5000 and FOR 1451, with the grants EB 555/3-1, WE 3081/15-2, EB 555/6-1, EB 555/6-2, ME 5474/1-1 and ME 5474/1-2; via the Priority Program 1374 'Biodiversity Exploratories', with the grants BL 960/8-5 and WE 3081/21-5; and via the iDiv (DFG–FZT 118, 202548816), which funded B.W. The Jena Experiment further received support from the Friedrich Schiller University Jena and the Max Planck Institute for Biogeochemistry. Field work permits were issued by the responsible state environmental offices of Baden-Württemberg, Thüringen and Brandenburg.

## Author contributions

A.E. conceived the idea of this study. A.E., E.L., S.T.M., H.S., M.S., J.H., C.R., O.S. and B.W. developed the concept of the study. R.A., N.B., M.B., A.E., L.H., M.S. and W.W.W. collected the data. B.W. analysed the data with support from M.B., A.E., E.L., S.T.M., H.S. and M.S. B.W. wrote the first paper draft. B.W. led the finalization and revisions of the paper with the help of all authors.

## Funding

## Competing interests

The authors declare no competing interests.

## Additional information

**Extended data** is available for this paper at https://doi.org/10.1038/s41559-025-02909-y.

**Correspondence and requests for materials** should be addressed to Benjamin Wildermuth.

¹Institute of Biodiversity, Ecology and Evolution, University of Jena, Jena, Germany. ²German Centre for Integrative Biodiversity Research (iDiv) Halle-Jena-Leipzig, Leipzig, Germany. ³Department of Biology, Duffy Science Centre, University of Prince Edward Island, Charlottetown, Prince Edward Island, Canada. ⁴Canadian Centre for Climate Change and Adaptation, University of Prince Edward Island, St. Peter's Bay, Prince Edward Island, Canada. ⁵School of Climate Change and Adaptation, University of Prince Edward Island, Charlottetown, Prince Edward Island, Canada. ⁶Terrestrial Ecology Research Group, Department of Life Science Systems, School of Life Sciences, Technical University of Munich, Freising, Germany. ⁷Ecological Networks, Technical University of Darmstadt, Darmstadt, Germany. ⁸Institute of Ecology, Leuphana University of Lüneburg, Lüneburg, Germany. ⁹Laboratoire d'Inventaire Forestier, ENSG, IGN, Université Gustave Eiffel, Nancy, France. ¹⁰Institute of Biology, Leipzig University, Leipzig, Germany. ¹¹Department of Physiological Diversity, Helmholtz Centre for Environmental Research - UFZ, Leipzig, Germany. ¹²Department of Community Ecology, Helmholtz Centre for Environmental Research - UFZ, Halle, Germany. ✉e-mail: bmwildermuth6@gmail.com

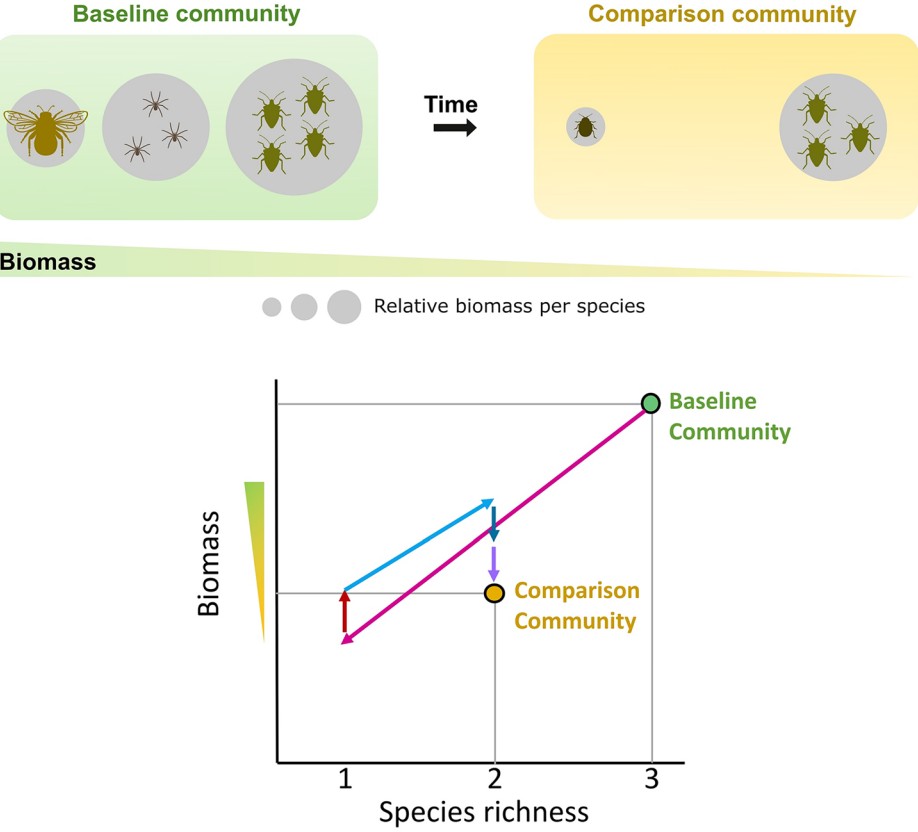

The Ecological **Price equation** (5 parts):
- **species richness** losses/gains (assuming average biomass)
- **species identity** losses/gains (correcting for species with non-average biomass)
- **abundance** changes in persisting species

**Extended Data Fig. 1 | Example visualization of temporal changes in arthropod community assembly and associated biomass between a baseline community and a comparison community using vectors (arrows).** The vectors are colored according to the info box. The vectors of community assembly components of biomass change are displayed in the order of 1) species richness loss, 2) species identity loss, 3) species richness gain, 4) species identity gain, and 5) abundance change of persisting species. The vector of species richness losses assumes that the two species lost from the baseline community have average biomass relative to the complete baseline community. However, since they have a below-average total biomass, the vector of species identity losses corrects for that. The vector of species richness gains assumes that the one species gained in the comparison community has average biomass relative to the complete comparison community. However, since it has a below-average total biomass, the vector of species identity gains corrects for that. Finally, the vector of abundance changes in persisting species reflects on the declining abundance and biomass of the one persisting species. Credit: arthropod icons, Gabriele Rada/iDiv.

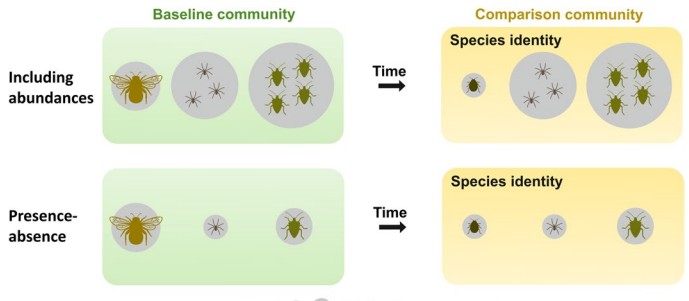

**Extended Data Fig. 2 | Illustration of the contrasting patterns of species identity effects on associated biomass change when analyzing total or individual biomass per species.** In the case of the total biomass per species (including abundances), the lost species from the baseline community has below-average biomass compared to the entire baseline community. In terms of individual biomass (presence-absence), however, the lost species from the baseline community has above-average biomass. For the gained species, this example shows less contrasting effects, yet also strong differences in the magnitude of the species identity effect compared to the entire comparison community. Credit: arthropod icons, Gabriele Rada/iDiv.

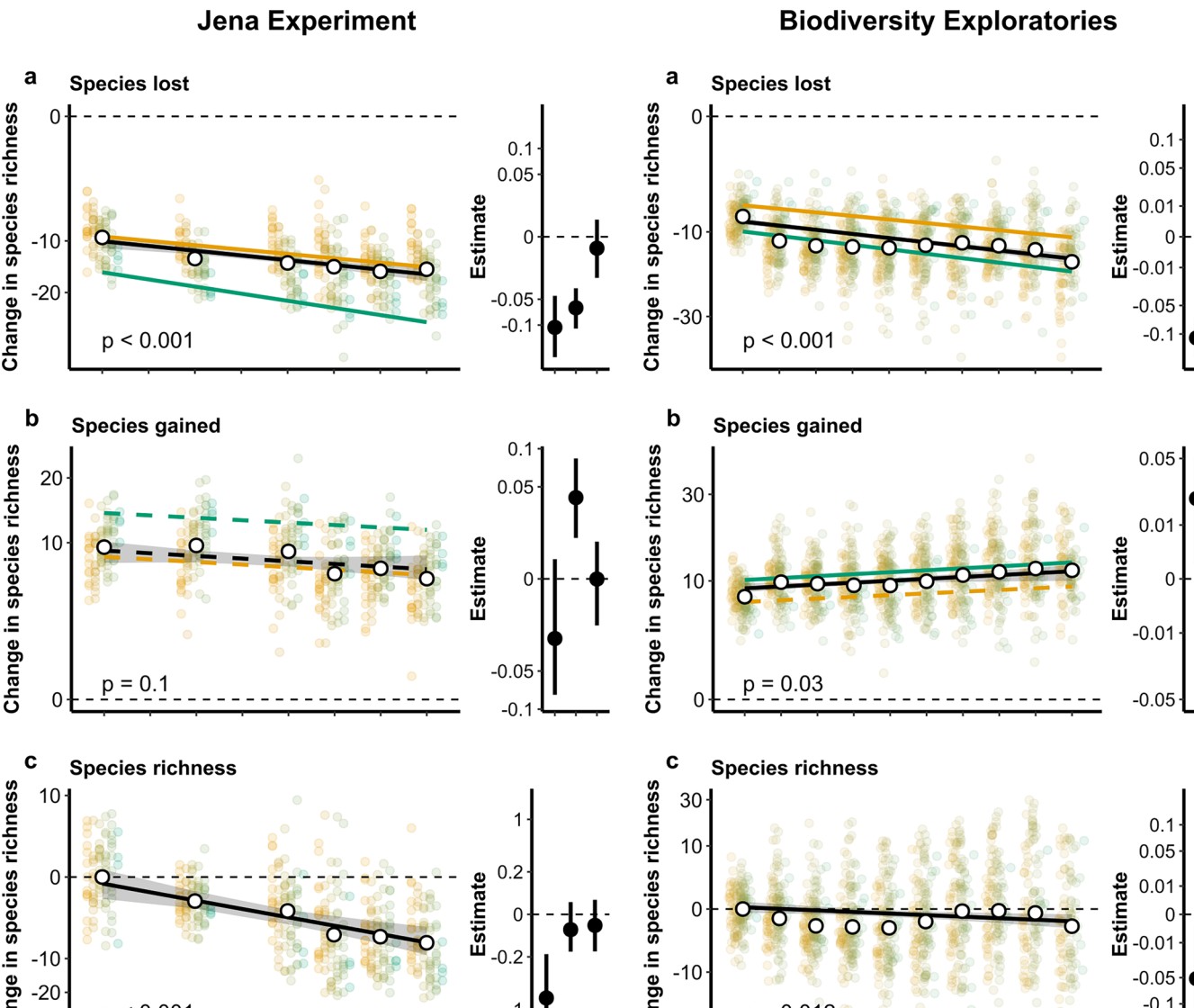

**Extended Data Fig. 3 | Arthropod species turnover is higher in plots of high plant species richness (PSR) and low land-use intensity (LUI), while overall species richness declines.** All panels are based on replicate-level (n = 160) median point estimates and predictions from 1,000 linear mixed-effects models drawing from shuffled data subsets, avoiding the reuse of sampling events in multiple pairwise comparisons. The main panels show temporal species turnover, as **a)** species lost, **b)** species gained and **c)** overall species richness change. Year 0 represents species turnover within years, between replicates (control). Mean species turnover values per replicate per plot are shown as colored dots along a PSR and LUI gradient (legend), white dots show mean values across all plots (n = 80). Solid regression lines indicate significant relationships (p < 0.05), dashed lines indicate non-significant relationships (p ≥ 0.1); exact two-sided p-values for the main effect of time are provided in the panels. Shaded areas around the black main effect line represent 95% confidence intervals. Significant effects of PSR and LUI on arthropod species turnover are shown in the main panels, colored along the PSR/LUI gradient (legend). Insets show median point estimates with 95% confidence intervals (error bars) for the effects of time and PSR/LUI on species turnover individually and in interaction (x).

# Jena Experiment

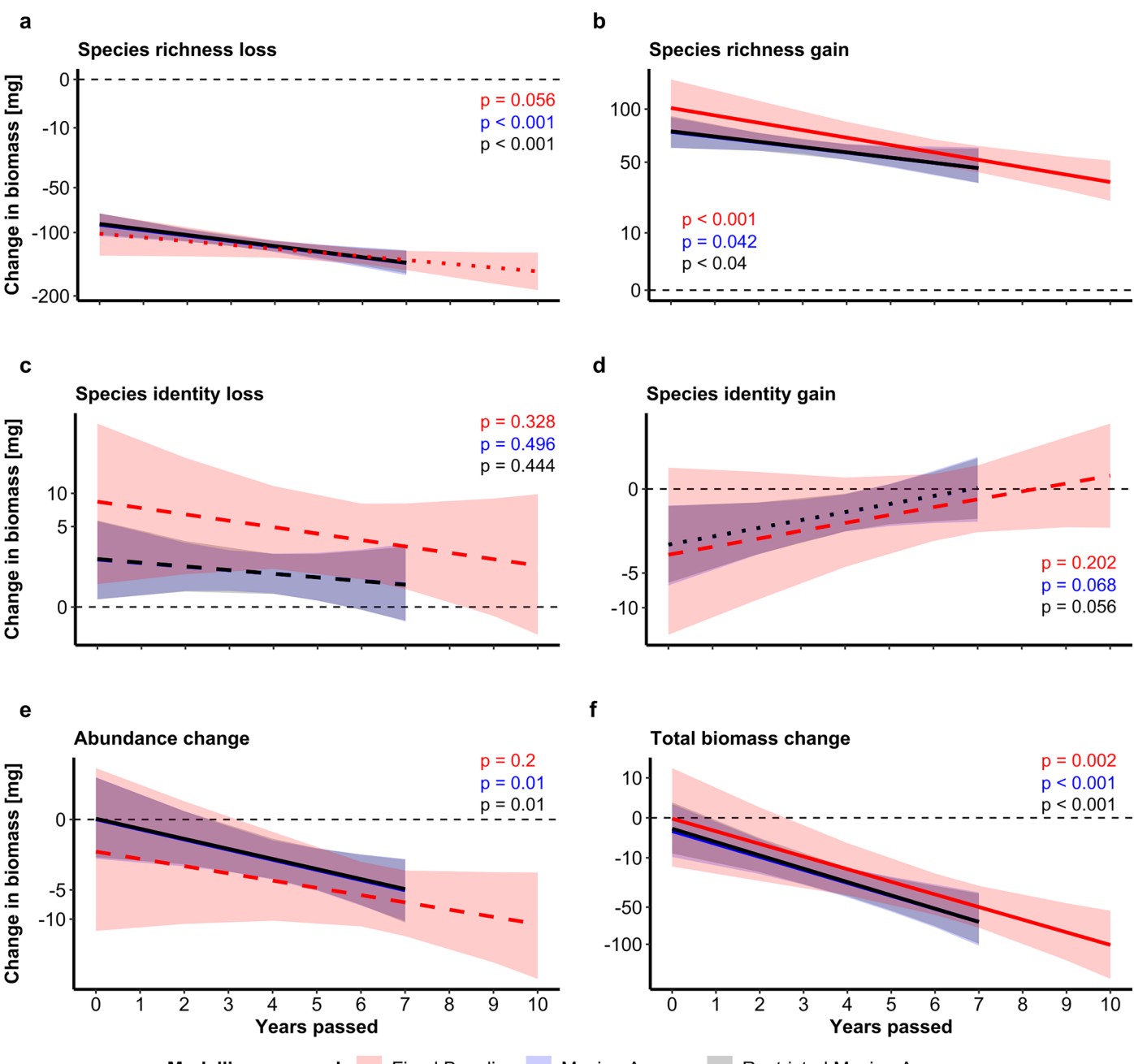

**Extended Data Fig. 4 | Different modelling approaches yield similar results of temporal biomass change in the Jena Experiment.** All panels are based on replicate-level (n = 160) median point estimates and predictions from 1,000 linear mixed-effects models drawing from shuffled data subsets, avoiding the reuse of sampling events in multiple pairwise comparisons. Each panel compares predicted regression lines for temporal biomass change according to the different modelling approaches: Fixed baseline comparison (2010 as the baseline; red), moving average comparison (2010-2019 as baseline; blue) and restricted moving average comparison (2010-2014 as baseline; black; used in

the main manuscript). Temporal biomass change was associated with **a**) species richness loss, **b**) species richness gain, **c**) species identity loss, **d**) species identity gain, and **e**) abundance change of persisting species (see Fig. 2 for detailed explanations). **f**) Total biomass change without partitioning. Solid lines indicate p < 0.05 (significant), dotted lines indicate p < 0.1 (marginally significant), dashed lines indicate non-significant relationships (p ≥ 0.1); exact two-sided p-values are provided in the panels. Shaded areas around the regression line represent 95% confidence intervals.

# Biodiversity Exploratories

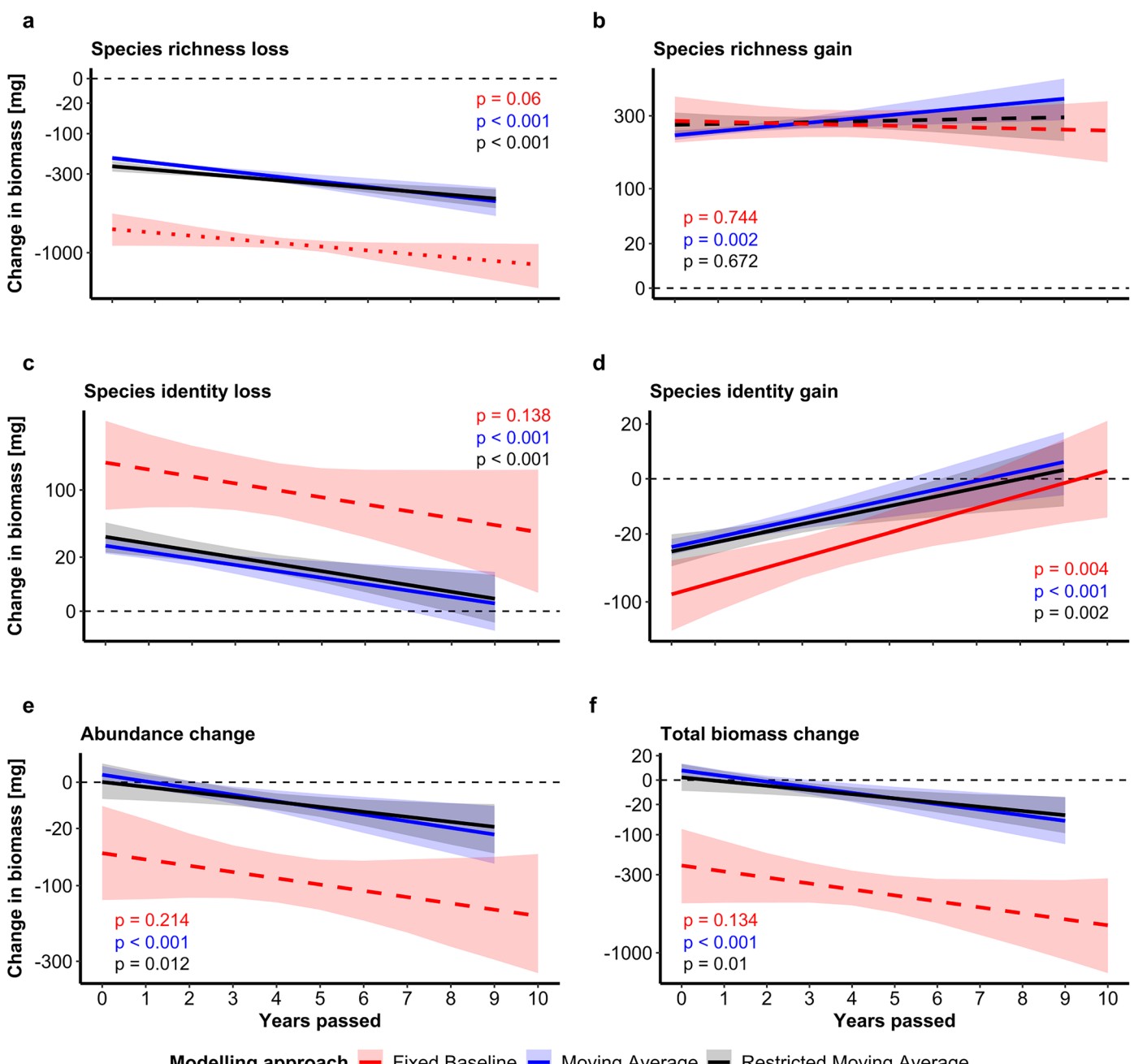

**Extended Data Fig. 5 | Different modelling approaches yield slightly diverging results of temporal biomass change in the Biodiversity Exploratories.** All panels are based on replicate-level (n = 300) median point estimates and predictions from 1,000 linear mixed-effects models drawing from shuffled data subsets, avoiding the reuse of sampling events in multiple pairwise comparisons. Each panel compares predicted regression lines for temporal biomass change according to the different modelling approaches: Fixed baseline comparison (2008 as the baseline; red), moving average comparison (2008-2017 as baseline; blue) and restricted moving average comparison (2008-2012 as baseline; black; used in the main manuscript). Temporal biomass change was associated with **a**) species richness loss, **b**) species richness gain, **c**) species identity loss, **d**) species identity gain, and **e**) abundance change of persisting species (see Fig. 2 for detailed explanations). **f**) Total biomass change without partitioning. Solid lines indicate p < 0.05 (significant), dotted lines indicate p < 0.1 (marginally significant), dashed lines indicate non-significant relationships (p ≥ 0.1); exact two-sided p-values are provided in the panels. Shaded areas around the regression line represent 95% confidence intervals. See Supplementary Note 1 for further interpretations.

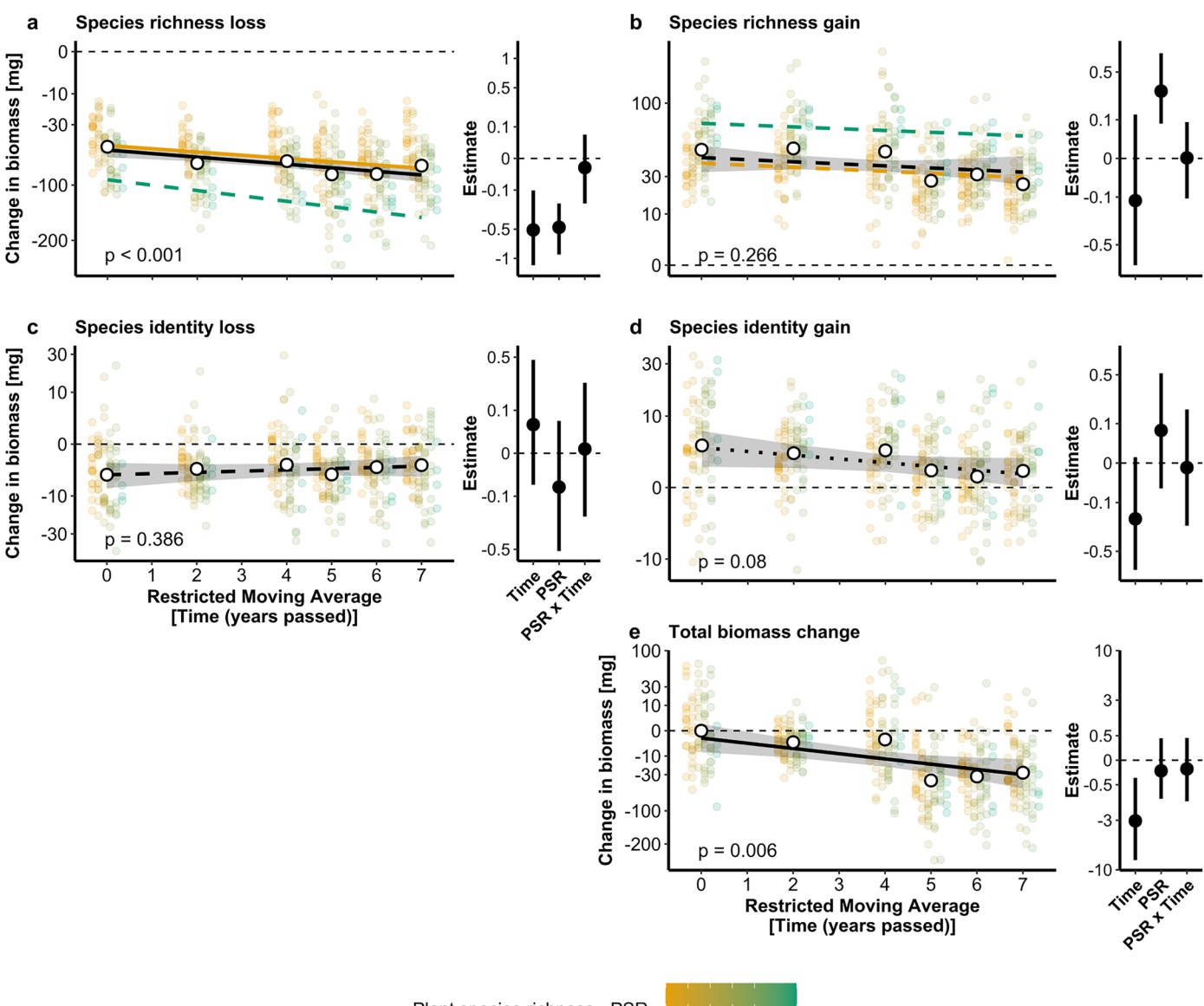

## Jena Experiment - Presence-absence

**Extended Data Fig. 6 | Biomass change associated with large species stagnates or declines over time while biomass loss associated with species loss increases in the Jena Experiment.** All panels are based on replicate-level (n = 160) median point estimates and predictions from 1,000 linear mixed-effects models drawing from shuffled data subsets, avoiding the reuse of sampling events in multiple pairwise comparisons. The main panels show temporal biomass change without abundance per replicate per plot (restricted moving average) associated with **a**) species richness loss, **b**) species richness gain, **c**) species identity loss, and **d**) species identity gain (see Fig. 2 for detailed explanations). **e**) Total biomass change without partitioning. Year 0 represents biomass change within years, between replicates (control). Mean biomass change values per replicate per plot are shown as colored dots along a plant species richness (PSR) gradient (legend), white dots show mean values across all plots (n = 80). Solid regression lines indicate significant relationships (p < 0.05), dotted lines indicate marginally significant relationships (p < 0.1), dashed lines indicate non-significant relationships (p ≥ 0.1); exact two-sided p-values for the main effect of time are provided in the panels. Shaded areas around the black main effect line represent 95% confidence intervals. Significant effects of PSR on arthropod biomass change are shown in the main panels, colored along the PSR gradient (legend). Insets show median point estimates with 95% confidence intervals (error bars) for the effects of time and PSR on biomass change individually and in interaction (x).

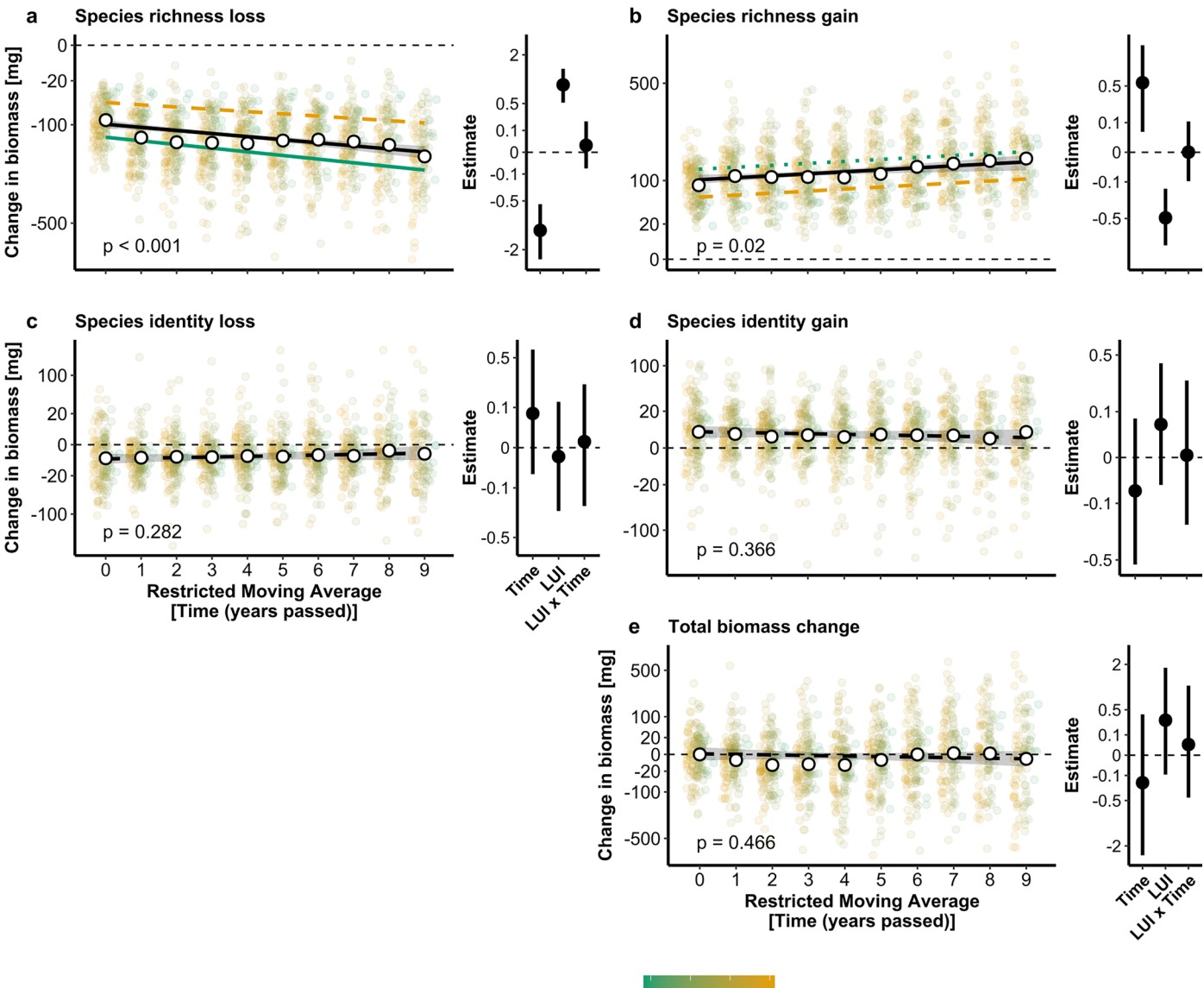

**Biodiversity Exploratories - Presence-absence**

**Extended Data Fig. 7 | Biomass change associated with large species stagnates over time while biomass loss associated with species loss increases in the Biodiversity Exploratories.** All panels are based on replicate-level (n = 300) median point estimates and predictions from 1,000 linear mixed-effects models drawing from shuffled data subsets, avoiding the reuse of sampling events in multiple pairwise comparisons. The main panels show temporal biomass change without abundance per replicate per plot (restricted moving average) associated with **a**) species richness loss, **b**) species richness gain, **c**) species identity loss, and **d**) species identity gain (see Fig. 2 for detailed explanations). **e**) Total biomass change without partitioning. Year 0 represents biomass change within years, between replicates (control). Mean biomass change values per replicate per plot

are shown as colored dots along a land-use intensity (LUI) gradient (legend), white dots show mean values across all plots (n = 150). Solid regression lines indicate significant relationships (p < 0.05), dotted lines indicate marginally significant relationships (p < 0.1), dashed lines indicate non-significant relationships (p ≥ 0.1); exact two-sided p-values for the main effect of time are provided in the panels. Shaded areas around the black main effect line represent 95% confidence intervals. Significant effects of LUI on arthropod biomass change are shown in the main panels, colored along the LUI gradient (legend). Insets show median point estimates with 95% confidence intervals (error bars) for the effects of time and LUI on biomass change individually and in interaction (x).

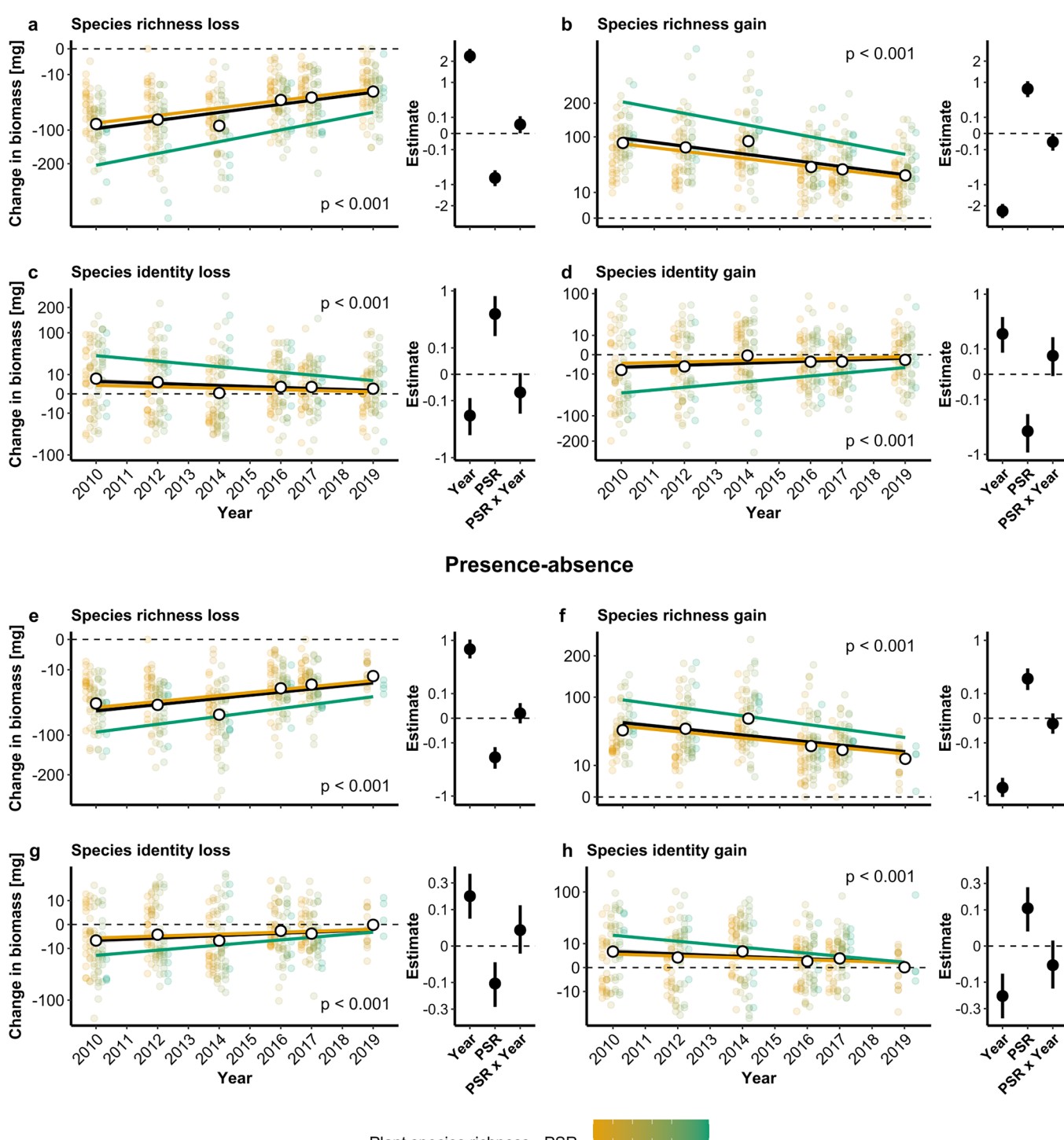

**Extended Data Fig. 8 | Species turnover among replicates, especially of rare species (above-average total biomass, below-average individual biomass), declines over time in the Jena Experiment.** All panels are based on replicate-level (n = 160) median point estimates and predictions from 1,000 linear mixed-effects models drawing from shuffled data subsets, avoiding the reuse of sampling events in multiple pairwise comparisons. The main panels show spatial (within plot, between replicates) biomass change with abundance (**a-d**) and without abundance (**e-h**) per replicate per plot over the years associated with **a,e**) species richness loss, **b,f**) species richness gain, **c,g**) species identity loss, and **d,h**) species identity gain (see Fig. 2 for detailed explanations). Mean biomass change values per replicate per plot are shown as colored dots along a plant species richness (PSR) gradient (legend), white dots show mean values across all plots (n = 80). Solid regression lines indicate significant relationships (p < 0.05); exact two-sided p-values for the main effect of year are provided in the panels. Shaded areas around the black main effect line represent 95% confidence intervals. Significant effects of PSR on arthropod biomass change are shown in the main panels, colored along the PSR gradient (legend). Insets show median point estimates with 95% confidence intervals (error bars) for the effects of year and PSR on biomass change individually and in interaction (x).

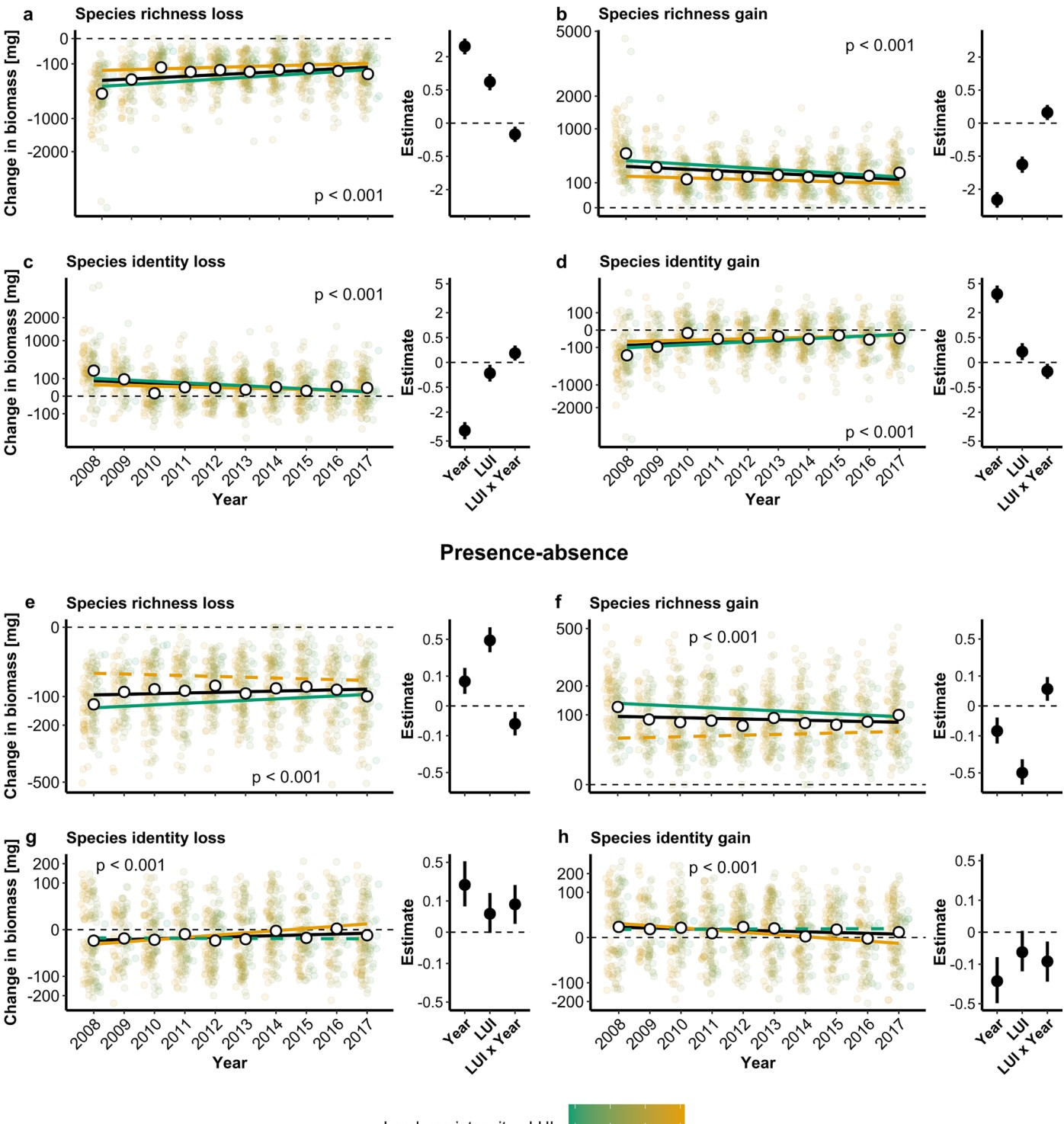

**Extended Data Fig. 9 | See next page for caption.**

**Extended Data Fig. 9 | Species turnover among replicates, especially of rare species (above-average total biomass, below-average individual biomass), declines over time in the Biodiversity Exploratories.** All panels are based on replicate-level (n = 300) median point estimates and predictions from 1,000 linear mixed-effects models drawing from shuffled data subsets, avoiding the reuse of sampling events in multiple pairwise comparisons. The main panels show spatial (within plot, between replicates) biomass change with abundance (**a-d**) and without abundance (**e-h**) per replicate per plot over the years associated with **a,e**) species richness loss, **b,f**) species richness gain, **c,g**) species identity loss, and **d,h**) species identity gain (see Fig. 2 for detailed explanations).

Mean biomass change values per replicate per plot are shown as colored dots along a land-use intensity (LUI) gradient (legend), white dots show mean values across all plots (n = 150). Solid regression lines indicate significant relationships (p < 0.05), dashed lines indicate non-significant relationships (p ≥ 0.1); exact two-sided p-values for the main effect of year are provided in the panels. Shaded areas around the black main effect line represent 95% confidence intervals. Significant effects of LUI on arthropod biomass change are shown in the main panels, colored along the LUI gradient (legend). Insets show median point estimates with 95% confidence intervals (error bars) for the effects of year and LUI on biomass change individually and in interaction (x).

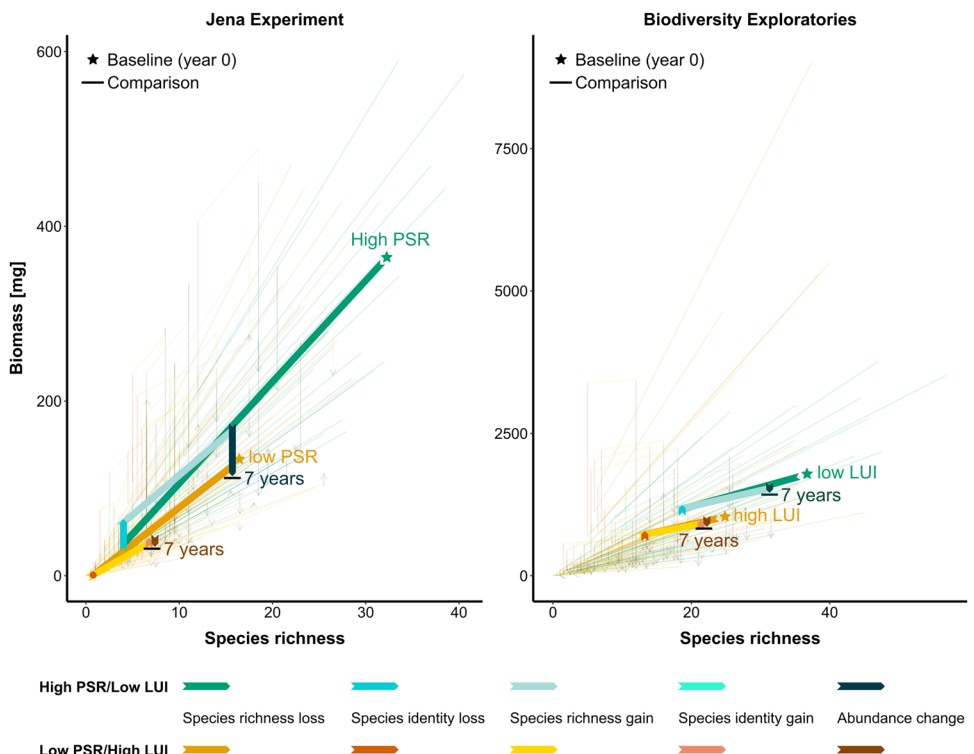

**Extended Data Fig. 10 | Visualization of the raw data spread underlying the vector (arrow) illustration of all 5 community assembly components associated with biomass loss.** Modelled change (thick vectors; restricted moving average) and raw data change (thin vectors) of arthropod biomass (y axis) and species richness (x axis) per replicate per plot after 7 years in dependence on plant species richness (PSR) and land-use intensity (LUI). Starting at the average community biomass and species richness value in the first year (Baseline), the community assembly components of biomass change are displayed as vectors (arrows) in the order of 1) species richness loss, 2) species identity loss, 3) species richness gain, 4) species identity gain, and 5) abundance change of persisting

species (see Fig. 2 for detailed explanations), reaching the predicted comparison community values of absolute biomass and species richness after 7 years. Most arthropod biomass change is associated with species richness change, but the vertical vectors of species identity and abundance change show that species richness change alone does not explain all biomass change. Modelled vectors are based on median predictions from 1,000 linear mixed-effects models (see methods and Tables S22, S23). Vectors for high PSR (60 plant species) and low LUI (0.5) plots are colored in green/blue, vectors for low PSR (monoculture) and high LUI (3.5) plots are colored in beige/brown (see legend).

# Reporting Summary

## Statistics

For all statistical analyses, confirm that the following items are present in the figure legend, table legend, main text, or Methods section.

| n/a | Confirmed | |
|---|---|---|
| ☐ | ☒ | The exact sample size (*n*) for each experimental group/condition, given as a discrete number and unit of measurement |
| ☐ | ☒ | A statement on whether measurements were taken from distinct samples or whether the same sample was measured repeatedly |
| ☐ | ☒ | The statistical test(s) used AND whether they are one- or two-sided<br>*Only common tests should be described solely by name; describe more complex techniques in the Methods section.* |
| ☐ | ☒ | A description of all covariates tested |
| ☐ | ☒ | A description of any assumptions or corrections, such as tests of normality and adjustment for multiple comparisons |
| ☐ | ☒ | A full description of the statistical parameters including central tendency (e.g. means) or other basic estimates (e.g. regression coefficient) AND variation (e.g. standard deviation) or associated estimates of uncertainty (e.g. confidence intervals) |
| ☐ | ☒ | For null hypothesis testing, the test statistic (e.g. *F*, *t*, *r*) with confidence intervals, effect sizes, degrees of freedom and *P* value noted<br>*Give P values as exact values whenever suitable.* |
| ☒ | ☐ | For Bayesian analysis, information on the choice of priors and Markov chain Monte Carlo settings |
| ☒ | ☐ | For hierarchical and complex designs, identification of the appropriate level for tests and full reporting of outcomes |
| ☐ | ☒ | Estimates of effect sizes (e.g. Cohen's *d*, Pearson's *r*), indicating how they were calculated |

*Our web collection on statistics for biologists contains articles on many of the points above.*

## Software and code

Policy information about availability of computer code

| Data collection | - |
|---|---|
| Data analysis | All analyzes were conducted in R 4.2.2 and newer versions. All relevant code for the main analysis can be found at https://github.com/BWildermuth/Model_every_species_counts |

For manuscripts utilizing custom algorithms or software that are central to the research but not yet described in published literature, software must be made available to editors and reviewers. We strongly encourage code deposition in a community repository (e.g. GitHub). See the Nature Portfolio guidelines for submitting code & software for further information.

## Data

Policy information about availability of data

All manuscripts must include a data availability statement. This statement should provide the following information, where applicable:
- Accession codes, unique identifiers, or web links for publicly available datasets
- A description of any restrictions on data availability
- For clinical datasets or third party data, please ensure that the statement adheres to our policy

This work is based on data from the Jena Experiment (DFG research unit FOR 5000 and FOR 1451) and the Biodiversity Exploratories (DFG Priority Program 1374). The analyzed data from the Jena Experiment are available on https://jexis.idiv.de/ under the identifiers 747 (raw data, available upon request due to ongoing

# Research involving human participants, their data, or biological material

Policy information about studies with human participants or human data. See also policy information about sex, gender (identity/presentation), and sexual orientation and race, ethnicity and racism.

| | |
|---|---|
| Reporting on sex and gender | *Use the terms sex (biological attribute) and gender (shaped by social and cultural circumstances) carefully in order to avoid confusing both terms. Indicate if findings apply to only one sex or gender; describe whether sex and gender were considered in study design; whether sex and/or gender was determined based on self-reporting or assigned and methods used. Provide in the source data disaggregated sex and gender data, where this information has been collected, and if consent has been obtained for sharing of individual-level data; provide overall numbers in this Reporting Summary. Please state if this information has not been collected. Report sex- and gender-based analyses where performed, justify reasons for lack of sex- and gender-based analysis.* |
| Reporting on race, ethnicity, or other socially relevant groupings | *Please specify the socially constructed or socially relevant categorization variable(s) used in your manuscript and explain why they were used. Please note that such variables should not be used as proxies for other socially constructed/relevant variables (for example, race or ethnicity should not be used as a proxy for socioeconomic status). Provide clear definitions of the relevant terms used, how they were provided (by the participants/respondents, the researchers, or third parties), and the method(s) used to classify people into the different categories (e.g. self-report, census or administrative data, social media data, etc.) Please provide details about how you controlled for confounding variables in your analyses.* |
| Population characteristics | *Describe the covariate-relevant population characteristics of the human research participants (e.g. age, genotypic information, past and current diagnosis and treatment categories). If you filled out the behavioural & social sciences study design questions and have nothing to add here, write "See above."* |
| Recruitment | *Describe how participants were recruited. Outline any potential self-selection bias or other biases that may be present and how these are likely to impact results.* |
| Ethics oversight | *Identify the organization(s) that approved the study protocol.* |

Note that full information on the approval of the study protocol must also be provided in the manuscript.

# Field-specific reporting

Please select the one below that is the best fit for your research. If you are not sure, read the appropriate sections before making your selection.

☐ Life sciences ☐ Behavioural & social sciences ☒ Ecological, evolutionary & environmental sciences

For a reference copy of the document with all sections, see nature.com/documents/nr-reporting-summary-flat.pdf

# Ecological, evolutionary & environmental sciences study design

All studies must disclose on these points even when the disclosure is negative.

| | |
|---|---|
| Study description | Temporal analysis (11 years) of arthropod biomass change in the Jena Experiment and the Biodiversity Exploratories. The former is a replicated Biodiversity Experiment with 80 plots of manipulated plant diversity grouped into 6 levels of diversity. The latter is a spatially replicated network of land-use observatories, spanning a range of land-use intensities. |
| Research sample | Arthropods sampled via suction sampling (Jena Experiment) and sweep netting (Biodiversity Exploratories). The samples represent the community level. |
| Sampling strategy | Suction sampling of two locations in each plot (Jena Experiment) and sweep netting along 150 m transects (Biodiversity Exploratories) sufficently sample the present arthropod community per plot. |
| Data collection | Samples were taken by members of the core arthropod projects in both research programs; see methods for detailed descriptions. |
| Timing and spatial scale | Jena Experiment: plot size = 5 x 6 m, sampling: 2010, 2012, 2014, 2016, 2017, 2019, 2020<br>Biodiversity Exploratories: plot size = 50 x 50 m, sampling: 2008-2018 |
| Data exclusions | In the Jena Experiment, the 2018 sampling had to be excluded due to incomplete sampling (infrastructure problems in 2018) |
| Reproducibility | The analyzed data are not experimental. (In the Jena Experiment, each plant diversity level is replicated) |
| Randomization | In the Jena Experiment, plant diversity is controlled in 6 levels, in the Biodiversity Exploratories, land-use intensity is measured yearly |
| Blinding | Blind sampling of arthropods is not possible, but sampling was strictly standardized |

Did the study involve field work?  ☒ Yes  ☐ No

## Field work, collection and transport

| | |
|---|---|
| Field conditions | Dry and windless, morning dew had dried |
| Location | Germany, see methods |
| Access & import/export | Collaboration with the administration of the Hainich national park, the UNESCO Biosphere Reserve Swabian Alb and the UNESCO Biosphere Reserve Schorfheide-Chorin. |
| Disturbance | - |

# Reporting for specific materials, systems and methods

We require information from authors about some types of materials, experimental systems and methods used in many studies. Here, indicate whether each material, system or method listed is relevant to your study. If you are not sure if a list item applies to your research, read the appropriate section before selecting a response.

## Materials & experimental systems

| n/a | Involved in the study |
|---|---|
| ☒ | ☐ Antibodies |
| ☒ | ☐ Eukaryotic cell lines |
| ☒ | ☐ Palaeontology and archaeology |
| ☐ | ☒ Animals and other organisms |
| ☒ | ☐ Clinical data |
| ☒ | ☐ Dual use research of concern |
| ☒ | ☐ Plants |

## Methods

| n/a | Involved in the study |
|---|---|
| ☒ | ☐ ChIP-seq |
| ☒ | ☐ Flow cytometry |
| ☒ | ☐ MRI-based neuroimaging |

## Animals and other research organisms

Policy information about studies involving animals; ARRIVE guidelines recommended for reporting animal research, and Sex and Gender in Research

| | |
|---|---|
| Laboratory animals | - |
| Wild animals | Arthropods were sampled in ethanol |
| Reporting on sex | - |
| Field-collected samples | Arthropods were morphologically identified |
| Ethics oversight | Not necessary for arthropod samples |

Note that full information on the approval of the study protocol must also be provided in the manuscript.

## Plants

| | |
|---|---|
| Seed stocks | *Report on the source of all seed stocks or other plant material used. If applicable, state the seed stock centre and catalogue number. If plant specimens were collected from the field, describe the collection location, date and sampling procedures.* |
| Novel plant genotypes | *Describe the methods by which all novel plant genotypes were produced. This includes those generated by transgenic approaches, gene editing, chemical/radiation-based mutagenesis and hybridization. For transgenic lines, describe the transformation method, the number of independent lines analyzed and the generation upon which experiments were performed. For gene-edited lines, describe the editor used, the endogenous sequence targeted for editing, the targeting guide RNA sequence (if applicable) and how the editor was applied.* |
| Authentication | *Describe any authentication procedures for each seed stock used or novel genotype generated. Describe any experiments used to assess the effect of a mutation and, where applicable, how potential secondary effects (e.g. second site T-DNA insertions, mosiacism, off-target gene editing) were examined.* |

