## [Peer Review File · Nature Ecology & Evolution]

Arthropod species loss underpins biomass declines

Corresponding Author: Dr Benjamin Wildermuth

This manuscript has been previously reviewed at another journal. This document only contains information relating to versions considered at Nature Ecology & Evolution.

Version 0:

Decision Letter:

26th September 2025

Dear Benjamin,

Your revised manuscript "Every species counts: Arthropod species loss, but not their identity, underpins biomass declines" has been seen again by the original reviewers, whose comments are provided below. Based on their feedback, I am pleased to inform you that we decided in principle to publish your work in Nature Ecology & Evolution, pending minor revisions to satisfy the reviewers' final requests and to comply with some editorial points and our formatting guidelines.

We are now performing detailed checks on your paper and will send you a checklist detailing our editorial and formatting requirements in about 10-12 days (normally we try to do this faster but I'll be travelling next week). Please do not upload the final materials or make any revisions until you receive this additional information from us.

[redacted]

Reviewer #1 (Remarks to the Author):

I reviewed an earlier draft of this manuscript, and then concluded that the paper provided valuable insights and was worthy of eventual publication, although I had some specific qualms that I wished to see addressed (as did the other reviewers). Most of those points are substantially addressed in the current manuscript. The authors do a much better job of making it clear that the changes being described are inferences from community samples, and that as such claims about the dynamics of the actual communities themselves -- let alone about regional or global arthropod dynamics, need to be expressed with appropriate caveats. The paper as it stands is generally clear, well argued and well supported by two excellent datasets. It makes a strong contribution to an ongoing debate.

My only somewhat-major concern remaining at this point is the assertion (e.g. lines 234-5) that constant sampling effort, the restriction of sampling to "dry and windless" days (line 415) and the use of the "control" of examining turnover within-site between replicate samples solves the problem of pseudo-extinctions and immigrations. Constant effort sampling is commendable, but it doesn't equate to constant intensity sampling: subtle differences in weather or timing can translate to large shifts in insect catches. Restricting samples to "dry and windless" days (line 415) helps somewhat, but differences in temperature, humidity, and random events such as emergence or migration events can produce substantial stochasticity in insect activity and thus catch sizes, despite constant effort. Their reply to my earlier query about incorporating weather data into their analyses ("...our analysis is not aiming at the external drivers of arthropod declines") misses the point in this respect: CLIMATE may be a driver of insect decline, but WEATHER on the day of sampling is a sampling issue. I am also puzzled by how the restriction of analyses to species occurring in at least 10% or 30% of plots helps address this problem. The finding (lines 271-272) that ignoring rare species reduces the proportional effect of species richness change on biomass seems to confirm the reviewers' concern that much of the turnover may be down to poorly sampled species (although, to be fair, rare species are also mostly likely to have genuine local extinctions). Meanwhile, the fact that this decline in relative richness effects is associated with increasing relative contributions of species identity and abundance (lines 272-273) is surely unremarkable (as these are relative contributions). Nor is it surprising that removing the rarest detected species from consideration decreased the measured contribution or rare species turnover (277-278).

Some minor points:

Lines 68-69: LOCAL species richness misses homogenization effects (but these are reflected in coarser-scale richness).

87-89: Awkward sentence; re-phrase.

109: Awkward phrase; "into the underlying community (dis-)assembly PROCESSES"? (or components?)

179-182: I'm not convinced that 8 years (2002-2010) is sufficient tie to pay off the extinction debts of a 1 ha experimental site. The cited reference shows community change slowing somewhat, but it would likely continue for some time (although I'd expect longer in terms of colonisation credits than for extinction debts). Also: previousLY (line 180).

Figure 4 (and Line 209): This figure is nicely information-dense, and useful, although I didn't entirely understand it. For instance, what does statement "vectors for species identity and abundance change only expand along the Y axis" (lines 331-332) mean? Also, with two 11-year time series, it's unclear why the focus here should only be on 7 years of change. Presumably this is linked to the limits of the Jena moving average comparison (lines 484-5; but if so, this should be explicitly stated). Would it be possible to assess the full 11-year temporal dynamics using the fixed baseline time series (in supplemental materials)?

225-226: "rare species" are usually determined in terms of abundance. "below average total biomass" and "above average individual biomass" gets close to that, as you could back-calculate abundance by dividing total biomass by individual biomass. But presumably you have access to the actual abundance data, which would define species rarity much more precisely. Why not look at that?

229: diminishING?

304: Is "assure" the correct word? "be certain" or "assume" perhaps?

362: Is it really true that "communities ... homogenized almost completely in later years"? If so, it would be good to see that quantified in some manner (e.g. through between-sample community similarity or difference metrics)

468-469: Unclear what the meaning of "fallback to the 'Chao' estimator" means in this context. Fall back under what circumstances?

Reviewer #2 (Remarks to the Author):

Thank you for sending me an update on this manuscript. I am pleased to read that the Authors of this manuscript have made substantial progress. I am the original Referee 2 and as such I write this review mostly as if this manuscript were a resubmission to the same journal. I am glad that my comments have been helpful so far.

I do believe that the Editors have made the correct decision in transferring the manuscript to Nature Ecology & Evolution, which is of course another top-level publication. The subject matter, global arthropod declines, is of utmost importance and should be addressed in such top publications. This manuscript describes important and novel aspects of biodiversity change, from a good dataset, and thus should be published when finalized. The resampling remedy employed by the Authors does seem to have technically addressed the issue of pseudoreplication at the data modelling stage. I still believe that to keep reporting back-and-forth between the Jena and Biodiversity Exploratories datasets is a little limiting on the Reader's comprehension of an overall bigger picture, and that some of the analytical methods (which would probably show the same trends, only in a different format) would be a little easier to comprehend through Generalized Additive Mixed Modelling. This is a complex manuscript, with a lot of concepts and some concepts (eg Price equations) which are probably new to many Readers interested in arthropod declines and thus really need to be explained 1) in sufficient depth and 2) in constant relation to the actual subject focus of arthropod ecology and threats. Those are, however, mostly my subjective opinions on structure rather than a criticism of integrity. I have commented on the Authors' responses below. I have reduced my original comments to save space and maintain dialogue.

Thank you for sending me this interesting manuscript to review...

#3. Response: We thank the referee for acknowledging the high relevance of our work. The review provided has indeed helped immensely to improve the robustness of our findings.

Reviewer 2: I am happy to see your advancement.

Key results...

Validity...

#4. Response: We thank the referee for drawing our attention to this issue. We agree with the referee that better accounting for the pairwise comparison structure and resulting problems of pseudoreplication was needed. We are pleased to state that after adapting our analysis accordingly, the main results remain valid. Please find the detailed responses and edits in response #6.

#5. Response: Yes, we indeed tried to control for the pairwise comparisons and the multiple occurrences of each datapoint

with the referenced random effect structure. We agree that our approach did not tackle the issue of pseudo-inflation appropriately and reanalyzed everything with more appropriate methods, see response #6.

I suggest...

#6. Response: We are very thankful for the ideas provided on how to remedy the previous issues. As pointed out by the referee, option 1 is very close to our approach, and thus, we went ahead and subsampled our data without replacement 1,000 times while avoiding any double use of unique samples. We then derived median point estimates, predictions and 95% confidence intervals from the distribution of the 1,000 model coefficients as suggested by the referee. Avoiding loss of an overwhelming proportion of our datapoints in each iteration, we did not only resample each plot once, but sampled for as many plot/year combinations as possible while guaranteeing single use of unique samples. For instance, if a plot in a moving average analysis in the Biodiversity Exploratories was randomly subsampled for the comparison 2008 v 2010, these two years were added to a blacklist and further subsamples could only draw pairs from the remaining years (2009, 2011, ..., 2018), with an increasing blacklist and thus a limited number of total possible subsamples per plot. In that way, we ensured that no year could appear twice in two picked pairs and thus no data point was recycled in multiple pairwise comparisons. We kept plot, year.x and year.y as random effect because they could appear in multiple comparisons ('(1 | plot) + (1 | year.x) + (1 | year.y)'). Fortunately, the main findings remained valid, meaning that we could solidify them. Only few significances of declining absolute species identity values were lost due to the reduced number of data points used, but their relative contributions are still sharply declining. Just one thought about option 2: we are not sure if there is a way to use the autocorrelation structure as suggested, since we have multiple data points for each time point (pairwise comparison) at each plot due to the moving average analysis. The relevant method section now reads:

L493 (clean manuscript): Using linear mixed-effects models in the lme4 package⁷⁷ and linear models (base R; only used for fixed baseline comparisons), we fitted models for each replicate-level biomass and species richness component as response. We included the temporal differences including controls (between-replicates turnover, i.e. year 0) as fixed baseline comparison, moving average comparison and restricted moving average comparison in interaction with plant species richness (PSR; defined as the initially sown plant species richness; Jena Experiment) or land-use intensity (LUI; Biodiversity Exploratories) as scaled continuous fixed effects⁷⁸. We further added plant biomass change as predictor in a supplementary model of total arthropod biomass change in the Jena Experiment, accounting for possible effects of systematically declining plant biomass⁴³ (Table S9; data derived from ref^{56,79}). In additional sensitivity analyses, we excluded all control comparisons from the models, analyzing solely changes in Price components among increasing time spans (Supplementary Material 1, Fig S4, S5). Since our data structure compares each sample across multiple time spans, using all pairwise comparisons in one model would inflate the degrees of freedom, introducing pseudoreplication⁸⁰. Therefore, we randomly sampled 1,000 subsets of the pairwise comparisons, with the sample size determined by the maximum number of possible random pairs without reusing any plot/year combination. We then ran models for each subset. We included the sampling plot and each contributing sampling year of each temporal comparison (year baseline and year comparison) as random effects in the moving average mixed effects models because both were sampled multiple times across unique combinations. Since for the fixed baseline models, each plot was only picked once per resampling and the temporal difference was covered by the fixed effect (the baseline is fixed, therefore each temporal comparison consists of unique years), we used a simple linear model without random effects. We obtained point estimates for low and high PSR and LUI values using the "emtrends" function from the emmeans package⁸¹ and linear predictions via the "ggemmeans" function from the ggeffects package⁸². We derived median point estimates, predictions and 95% confidence intervals from the distribution of the 1,000 model coefficients and predictions^{80,83}. We further calculated two-sided p-values of point estimates as twice the proportion of point estimates falling on the less frequent side of zero⁸⁴. To estimate the combined contributions of i) species richness losses and gains, ii) species identity losses and gains and iii) all species turnover components to overall biomass change, we also derived the median and 95% confidence interval for their combined 1,000 predictions. Moreover, to analyze the temporal development of within-year between-replicate species turnover, we fitted a model exclusively for the controls across all years (see Discussion; Fig S8 - S11, Table S12 - S15). Although the pairwise comparisons in this model do not cover multiple time spans, pseudoreplication was still an issue as each sample was compared two-fold, once as baseline and once as comparison (replicate A vs B, and B vs A). Therefore, also this model was run across 1,000 subsets, sampling each plot/year combination once per iteration (either A vs B or B vs A were sampled) and plot was included as random effect in the model.

Reviewer 2: I believe that this does solve the issue of pseudoreplication at the modelling stage, and I was able to follow the updated Methods description. I still believe that the use of GAMMs would probably be easier to follow and therefore be more accessible to a broader readership, especially given the complexity in using ecological Price equations and two separate datasets. Again, though, I think this approach is probably overall accurate, just my opinion.

Modelling aside...

#7. Response: The main issue raised here (sample coverage; is it real turnover or just sampling noise) alarmed all three referees and, therefore, we implemented substantial changes to address the concern. Please refer to our extensive response to referee 1 (response #1; #1.4 and #1.5 for coverage & mobility). We want to add here, that grouping of plots is not advisable in our case because each plot has a unique portfolio of sown plant species in the Jena Experiment and also the plots in the Biodiversity Exploratories are unique. Grouping would thus reduce/remove replication within each level of plant species richness and land-use intensity, preventing statistical analyses of environmental factors, and mix up turnover patterns associated with specific plant species compositions or land-management practices.

Reviewer 2: Ok sure, it does seem irrelevant to group plots, I understand that decision. I would though suggest that getting into the mathematical quantification of coverage/entropy may be a bit of a rabbit hole – most estimators assume a closed system, and my previous points largely pertained to highly dispersive species (for example, adult beetles flying) around and unpredictably landing in small sampling sites. Therefore, I don't think those coverage values of ~90% are very meaningful,

as each small site is an open system which will experience many visitors. My advice would be to place emphasis on biological knowledge and admission of caveats on arthropod sampling given the sampling design and such arthropod behavior. I think arthropod scientists would generally accept that, and it would probably save you undue hassle in attempting to mathematically define coverage! The mobility part requires a brief description of what A-D and 0-1 mean as dispersal measures, as not all Readers will access those associated references.

One specific misunderstanding shall be clarified here though: the analyses with minimum thresholds of 10% and 30% occurrence per plot per year were supplementary analyses, investigating the robustness of our findings even when excluding very rare species including singletons, doubletons etc. Since all referees gave the feedback that this was not written clearly enough in the previous manuscript version, we compiled our major revisions in response #1. Please refer to response #1.2 in particular.

Reviewer 2: Yes, I understood these thresholds better when explained in terms of limiting the noise component of the equations, I can see how that translates. I think that also highlights the previous emphasis on modelling, perhaps with the omission of arthropod biology.

#7.1 We further appreciate the thoughtful concerns regarding abundant taxa such as aphids and ants because it alerted us about missing information in the manuscript: we indeed did not count and identify aphids and ants – they were entirely excluded from the analysis.

L429: Among all sampled arthropods, we analyzed three common arthropod taxa in both research programs: the highly diverse order of Coleoptera, the herbivore-dominated Hemiptera (excluding Sternorrhyncha), and the predatory Araneae. Hymenoptera (representing diverse feeding guilds) were analyzed exclusively in the Jena Experiment (excluding Formicidae), while Orthoptera (mostly herbivorous) were included only in the Biodiversity Exploratories. Overall, these groups can be adequately captured in grasslands with the respective methods^{8,55}, covering some of the most diverse arthropod taxa and multiple feeding guilds.

Reviewer 2: Glad to see this important information included.

Originality and Significance:

The results, if valid, might be considered original in that they seemingly contradict some other high-profile studies, notably the Van Klink et al. (2023) article in Nature which found that insect abundance losses were mostly due to reductions in abundance of common species rather than species loss. However, this analysis does seem to use a lot of the same data as Seibold et al. (2019), also Nature, who previously reported reduced arthropod species richness, abundance and biomass from German grasslands and identified the largest declines in rare species. This manuscript does build on Seibold et al. by further dissecting the biomass declines specifically from grasslands but probably shouldn't be considered of the same level of significance because many of the results mirror those previous findings.

#8. Response: We appreciate the comparisons to van Klink et al. (2023) and Seibold et al. (2019). We agree that our results need to be evaluated in light of their findings. Importantly, we add conceptual and methodological advances and novel insights by disentangling the contributions of community (dis-)assembly to arthropod biomass loss while systematically evaluating the impacts of plant species richness and land-use intensity on these processes. We can state for the first time that while rare species may show increased turnover and decline, common species play the key role for local biomass declines, especially once communities are getting smaller (which they do over time, but also with decreasing plant species richness and increasing land-use intensity). It is moreover notable that our moving average analysis shows that single years can have extreme impacts on general trend estimation when using a fixed baseline; in this case, the arthropod decline is estimated to be more moderate in our analysis than in the Seibold (2019) paper. Lastly, combining the high-quality data sets of the Jena Experiment and the Biodiversity Exploratories makes our study not only novel in terms of question, methodology and results, but also give the overall dataset an unprecedented dimension and significance. Please find relevant sections referencing the two studies in the following lines and refer to our conclusions section for further statements of significance: L168: These rates are lower than previously reported annual decline rates of approximately 7% for biomass and 3.8%-5.6% for species richness in both research programs^{8,43}. Our use of a restricted moving average approach was expected to yield lower estimates than simple time series due to averaging of systematic temporal trends within each moving window. Yet, smoothing extreme years makes detected trends more generalizable and robust⁵¹, reducing, but not removing strong impacts of single sampling years with e.g. extreme climatic conditions (see Supplementary Material 1 for further sensitivity analyses).

Supplementary Material 1: L38: In the Biodiversity Exploratories, model intercepts and predicted absolute values of biomass losses were much more extreme in the fixed baseline comparisons than in the restricted and unrestricted moving average analyses (Fig. S3). This reflects the weakness of fixed baseline analyses: all comparisons rely on the value of one single baseline (2008), which, in this case, was an exceptionally good year for arthropod abundances². Therefore, relative losses in the following years were much higher than in moving average analyses, where multiple baseline years were included. As discussed in the main manuscript, we conclude that restricted moving average comparisons yield more robust and generalizable trends.

L227: We suggest, since arthropod species in grasslands were previously shown to decline the fastest⁸, that their diminished contribution to species turnover and biomass change is a symptom of their decline.

L242: We thus add robust findings of temporally homogenizing biomass distributions in arthropod communities to previous reports on homogenizing taxonomic and functional diversity^{23,59}. Our analyses suggest that this is both driven by homogenizing abundance distributions, that is, declines of rare species^{8,31} and homogenizing size distributions, i.e. declines of large-bodied species^{21,28}. Losses of rare species with potentially unique and complementary functional profiles threaten ecosystem functioning and resilience^{30,34}. At the same time, the overwhelming contribution of species richness loss per se to declining arthropod biomass reveals the consequences of arthropod communities increasingly losing their common members^{10,21}. This may have further escalating negative consequences for multitrophic diversity and ecosystem

functioning^{7,10,15}.

L373: Our results on fluctuating arthropod biomass in the Biodiversity Exploratories partly support these findings, but the general species turnover and biomass trends emphasize ongoing arthropod community simplification^{8,23,59}, and that even common species may be increasingly under threat^{10,21,33}.

Reviewer 2: I think it important to note that occasional climatic occurrences do have biological meaning also beyond statistical noise, for example the the recent and already referenced Sharp et al. 2025 paper on El Nino (admittedly from the tropics, and different threats). It is undoubtedly difficult to separate the noise from the meaning. That doesn't invalidate your justified methods. I believe it just requires a sentence or two of written recognition.

The results of this manuscript are directly relevant to human-modified grasslands, potentially outside of Germany also, but are perhaps limited beyond that. In my opinion, that regional/habitat limitation is not really appreciated and requires addressing in the text and title. By comparison, Hallman et al. (2017; PLOS One) limit their conclusions on biomass declines from malaise traps in protected areas of Germany to 'the European landscape'. I also believe that presenting the Jena Experiment and Biodiversity Exploratories data separately throughout reduces the perceived impact of the findings. I am forced to keep thinking back to the differences between those studies, and thus the reported results are presented as quite site/region-specific.

#9. Response: Agreed, we added some discussion on the potential limitations in the text. To avoid limiting our findings too drastically, we want to emphasize that our study is identifying intrinsic community (dis-)assembly mechanisms of declining biomass, and not another study quantifying the overall decline. While our restricted moving average approach indicates that the overall decline may be more moderate than previously reported (it depends on the baseline), this is not our main finding. Our main finding is that in shrinking arthropod communities, biomass declines are primarily associated with species declines per se. This pattern is statistically the most likely finding and thus potentially not only transferrable to other arthropod communities, but to any biotic community undergoing declines (Hillebrand et al., 2018). Accordingly, also Hallmann and colleagues kept the title more general when dissecting intrinsic relationships of biomass and species richness within the decline ((Hallmann et al., 2021) "Insect biomass decline scaled to species diversity: General patterns derived from a hoverfly community"). We are not entirely sure we understood the second point made by the referee about separating the findings from the two datasets. According to referee 3, we now highlight the differences between the datasets, including their scale, when discussing the results. It is, as pointed out by referee 3, very reassuring and worth emphasizing that the results remain consistent among the two datasets (with minor differences). Find the adapted sections here:

L37: Synthesizing 11 years of data from a biodiversity experiment and from farmed grasslands in Central Europe across a gradient of plant species richness and land-use intensity, we show that arthropod biomass declines were predominantly (> 90%) linked to species richness losses.

L149: Analyzing a total of 239,690 arthropod individuals across 1,572 species (Tables S2, S3), we found an overall decline of arthropod biomass and species richness in Central European grasslands over time in both time series (Jena Experiment: ~5% yearly declines, Biodiversity Exploratories: ~ 0.5% yearly declines, but see refs. 8,43).

L359: Our study provides insights into temporal declines of local arthropod biomass in anthropogenic Central European grasslands within a limited and relatively recent time frame (2008-2020). Within that time, we detected a small, yet significant role of species identity in early years, with rare species disproportionately contributing to species turnover. However, communities shrank and homogenized almost completely in later years, with their biomass declines primarily associated with species richness declines—a pattern which was consistent among the different setups and scales of the Jena Experiment (experiment, small scale) and the Biodiversity Exploratories (real world, larger scale), and which likely transfers to many shrinking biotic communities³⁵. These are concerning findings, hinting at simplified biomass distributions and highly vulnerable arthropod communities in the face of ongoing anthropogenic global change; yet, in other ecosystems around the world, the mechanisms behind arthropod biomass declines may be different and similar studies are needed elsewhere.

Reviewer 2: I agree with Reviewer 3 in that the differences in scale required addressing, but that relates to my points that without some direct methodological integration of the two datasets, the Authors and Reader must always consider one and then the other. I don't believe that the two actually should be integrated, given the inherent differences in sampling, scale and coverage. I mean that that is a 'natural' limitation in this analytical design which probably, to some degree to be decided by the Editors, reduces the relative impact. What I mean is evident in the above text: "the different setups and scales of the Jena Experiment (experiment, small scale) and the Biodiversity Exploratories (real world, larger scale)". I do not disagree that species richness losses largely underpin biomass reduction. Your results certainly support that conclusion, and that conclusion is also in line with much broader understanding that arthropod species richness declines accompany vegetation homogenization. The clarifications that the results pertain to Central European grasslands improve the accuracy of the text.

Data and Methodology:

While I do have some questions about the data treatment (described earlier), the arthropod data has been collected as part of existing long-term projects and published in peer-reviewed studies. The arthropod collection methods are different: suction sampling for the Jena Experiment and sweep-netting for Biodiversity Exploratories. Beetles, true bugs and spiders were the collected via both methods, but presumably the different sampling methods still picked up different species in those three taxonomic orders. I would expect to see summaries of the taxa collected in the different experiments, otherwise it's impossible to fully assess how appropriate the data used and resulting conclusions are.

#10. Response: The referee is right that the species lists among the two datasets vary considerably, partly because of the sampling technique and partly because of the different types of grassland and regions. We believe that (i) the relative nature of the analysis (only samples obtained by the same sampling technique are compared) and (ii) the striking consistency of observed (relative) patterns despite different scale, different management, different regions and different sampling techniques among the two research platforms justify the differences in taxonomic coverage. However, it is informative to

provide full taxon summaries, which are now added to the appendix (Table S2, S3) and the raw data are provided.

Reviewer 2: I do agree with your points but going back to arthropod biology as well as experimental design, we must also recognize that responses can differ at the family and even order levels due to differences in physiology and life histories. Inclusion of the taxon summaries is certainly helpful for Readers to make their own informed decisions on the impact. Again, I think those interested in arthropod biodiversity declines are broadly accepting of these difficulties and they don't detract from your analyses, just need a little recognition in the text.

Appropriate Use of Statistics and Treatment of Uncertainties:

I find that some models should probably use a non-gaussian error family for many of the response variables, for example species loss and gain, where all the values are non-negative or non-positive, and there remains significant skew in the square-root transformed response data. Indeed, the left-hand panel of Figure 4 reveals that at least one model predicts negative species richness and negative biomass at low values of plant species richness (impossible, of course). This could be rectified by using generalized models with a log-link function but is secondary to my major statistical observation (earlier).

#11. Response: The referee is right about the impossibility of having negative species richness and biomass values in fig. 4. This mistake happened because the starting point of the vector sequence was chosen as an average value of species richness and biomass in the first 5 years of sampling. However, the modelling of the restricted moving average is slightly more complex than just assuming an average value of the first five years as baseline. To keep the purpose of fig. 4, which is to show not only the relative changes in biomass/species richness (in that case, the starting point would be zero for all comparisons), but also to give an idea of the absolute baseline values which differ vastly between PSR/LUI levels, we now use the average value of the first year as starting point for the vectors – which fixes the issue of impossible negative values. Regarding the model family, we ran all types of different transformations, model families and links (even extremely flexible families such as SHASH()), and indeed, the gaussian family in combination with square root transformed response data fits the best. Find the revised figure in L324

Reviewer 2: Glad to see this rectified and the models checked.

It is difficult to comment on the suitability of the Price equation, which is central to this manuscript as, although it appears to fit purpose the equation is never reported and little justification is given. Some of the results, derived from the Price equation, are not immediately intuitive, for example how can there be effects of species loss and gain after 0 years? (Figures 2 and 3). Land Use Intensity (LUI) is also not described at any point, but is referenced as 58.

#12. Response: We agree that more explanation concerning the Price equation was needed; the reader can furthermore find the equation in the appendix now. We extended multiple sections in the main manuscript to provide more background (see below). The effects at year = 0 are our control, that is the turnover among spatial replicates, indicating the amount of "noise" we can expect due to detection issues (no two samples are ever going to be exactly the same, even when sampled under the exact same conditions). We added further explanation about our control in the main, as it seems to have been insufficiently explained for all the referees. The description for how land-use intensity (LUI) was calculated is kept short, but we list the included components of fertilization, grazing and mowing and provide the reference for more detailed information. Find the sections on the Price equation and LUI in the following lines and please refer to response #1.2 for detailed sections on the control:

L104: Separately for each time series, we used the ecological Price equation to partition temporal changes in local arthropod community biomass into the contributing components of community dis-(assembly). The Price equation was originally developed for quantifying changing gene frequencies under natural selection⁴⁶. The ecological adaptation partitions changes in ecosystem functions^{25,26,47}, or in our case biomass^{48,49}, between two communities into the underlying community (dis-)assembly, separating effects of average species turnover (species richness) from non-average species turnover (species identity) and effects independent from species turnover (here: Abundance change). Specifically, the five components are: i+ii) species losses and gains assuming that all species undergoing turnover have average biomass relative to their respective communities (expected effect of species richness), iii+iv) the difference between the expected and observed biomass change associated with species turnover, i.e. the deviation of lost and gained species from the average biomass of their respective communities (species identity of lost and gained species), and v) changes in abundance of persisting species²⁵ (Fig 1, see Supplementary Material 1 for the mathematical equation).

Supplementary material 1: L9: The 5-part ecological Price equation partitions changes in ecosystem function, or in our case biomass (BM), between two communities into five components: Species richness loss and gain (SRE.L, SRE.G), species identity loss and gain (SIE.L, SIE.G) and context-dependent effects (CDE), which in our case is attributable to abundance changes of persisting species (ABU). These definitions follow ref1, except for renaming the fifth component from CDE to ABU. The following equations are modified after ref1:

$$\Delta BM = SRE.L + SRE.G + SIE.L + SIE.G + CDE$$

with (see table S1 for the definitions of the variables):

$$SRE.L = (s_c - s) \cdot \bar{z}$$

$$SRE.G = (s' - s_c) \cdot \bar{z}'$$

$$SIE.L = s_c \cdot (\bar{z}' - \bar{z})$$

$$SIE.G = [(s_c - s) \cdot (\bar{z}' - \bar{z})]$$

$$ABU = s_c \cdot (\bar{z}' - \bar{z})$$

Table S1 Definitions of variables used in the ecological 5-part Price equation modified after ref1

L127: In interaction with time, we also assessed the effects of plant diversity (Jena Experiment) and land-use intensity (Biodiversity Exploratories; based on mowing, grazing and fertilization⁵⁴) on arthropod responses.

L404: We calculated land-use intensity (LUI) values based on the combined intensities of fertilization, grazing and mowing following ref. 54.

Reviewer 2: Glad to see this information included. As another Reviewer also stated, the concept of Price equations will be

new to many interested in arthropod declines and so they must be generally understandable without reference outside of this manuscript. There still isn't really any information given on LUI. Why not just a single sentence for understanding outside of a modelling context, for example "value X = no fertilization, grazing or mowing and value y = grazing or mowing to a barren state.." ?

The results presented in terms of percentages are not presented with error ranges, so it is impossible to assess uncertainty on them and thus they are currently unconvincing.

#13. Response: We added the CI ranges accordingly (see the results and discussion section in the main manuscript)
Reviewer 2: Glad to see.

Conclusions:

The conclusions relate to human-modified grasslands, in Germany and probably the surrounding region, but should not be extrapolated beyond that. There is frequent reference to 'rare' species, eg lines 291-292 'rare species disproportionately contributing to species turnover'. Some of these conclusions, including this one, I don't buy and instead expect that the perceived 'turnover' in the data to represent under-sampling and mobile species at plot scale rather than true temporal population fluctuations. Overall, many of the conclusions require further justification.

#14. Response: As outlined in the detailed responses above (particularly responses #1 and #9), we now highlight the geographical/habitat limitations of our study and implemented major revisions concerning the justification of our methods, findings and therefore, our conclusions.

Reviewer 2: Yes, agreed.

Suggested Improvements:

My most major points in brief, after which much of the text and figures might change:

- 1) Take another look at the statistical modelling, as suggested.
- 2) Either address the caveats on sampling coverage as suggested by reporting species accumulation or similar (and hopefully confirming that the 'rare' species are in fact rare rather than just lesser sampled), or group plots somehow to reduce the potential under-sampling effect.

#15. Response: We once again thank the referee for the constructive review, we were happy to implement the suggested improvements (see the detailed responses above). Fortunately, our findings remained valid and are now solidified.
Reviewer 2: I believe the first point has been addressed. The Authors have also provided background information on their 10% and 30% thresholds, more as sensitivity analysis to noise rather than specifically ecosystem-rare species. This makes more sense in the context of their analysis. Given that, and seconding an earlier comment of mine, I wouldn't put too much emphasis on the mathematical quantification of coverage in this open system ('open' because of how small some of the sites are). Coverage could instead be discussed and justified in relation to the sampling methods and arthropod behaviors.

References:

The referencing appears correct and the relevant literature to the insect declines debate is cited.

Clarity and Context:

From the outset, reference to 'Species Identity' is confusing. To me, this implies perhaps ecological role or phylogenetic relatedness to other species. I would avoid this terminology completely, perhaps 'Species Traits' is more appropriate.

#16. Response: We understand that the term species identity can be used in many contexts. In the context of the ecological Price equation, we only have two choices: both the terms "Species identity" and "Species composition" are established terms for the third and fourth Price component. Since our analysis relies on the R package from (Bannar-Martin et al., 2018), we chose the definitions provided there, also considering that a more recent high-impact publication on biomass change and the Price equation used the same framing around species identity (Lefcheck et al., 2021). Please refer to response #12 for the detailed section in the introduction and find changes in the caption to fig 1 in the following lines:

L135: The relative biomass per species is indicated by the size of the grey circles. Changes in the total community biomass can be associated with changes in a) species richness, b) species identity, and c) abundance of persisting species. The species richness component assumes an equal (average) contribution of all species to community biomass. However, species under turnover may have non-average biomass (b), in which case their identity must be considered. Consequently, the species identity component reflects the difference between the biomass change expected from the species richness component and the actual observed biomass change associated with species turnover.

Reviewer 2: I understand that, but this is another case where the decisions made are based on Price equations, rather than arthropod ecology and associated background knowledge. For example, in the context of the title where 'identity' is used without reference to Price equations, the word to me implies taxonomic identity.,

In my opinion, the grey weight shapes in Figure 1 don't make immediate sense. Perhaps the grey shapes could be replaced with simple circles or similar, with a key indicating discrete low/mid/high biomass?

#17. Response: We agree that circles are less confusing than the weight shapes. We updated the graph and we state that the size is equivalent to the relative biomass in the caption (response #16).

Reviewer 2: Agreed, improved.

The grey points at time = 0 in Figures 2 and 3 are confusing because it seems that they do not have PSR/LUI data

associated. I would just colour all points, including these, according to the same orange-green gradient.

#18. Response: The referee is right about the ambiguous nature of coloring the control in grey: it highlights that it is our control, not a temporal comparison - but we lose the information on PSR/LUI. We were hesitating about this decision and now we are happy to adapt it according to the suggestion. All data points are now colored.

Reviewer 2: Agreed, improved.

I honestly just don't understand Figure 4, after having re-examined it several times. I think either a different method of presenting this data or a clearer description of these vectors is beneficial. The pairing of PSR and LUI categories is particularly confusing, as are the missing arrows of certain colours.

#19. Response: Yes, figure 4 is always the tricky bit about Price equation papers. We do believe that the vector graph is the best way to show all Price components in one graph (see also (Bannar-Martin et al., 2018; Hogan et al., 2023; Ladouceur et al., 2022)), but we agree that the graph should always show each color, be it as a dot. We further modified the graph as such that labelling per PSR/LUI category shows the respective colors and that the arrows have a direction indicated. Also, the caption could use clarification. It now reads:

L324: FIGURE 4 Losses and gains of species with non-average total biomass (species identity) contribute disproportionately to arthropod biomass change under high plant species richness (PSR; Jena Experiment) and low land-use intensity (LUI; Biodiversity Exploratories). Modelled change of arthropod biomass per replicate per plot after 2 and 7 years (restricted moving average), in dependence of PSR and LUI. Starting at the average biomass value in the first year (indicated with a star symbol labelled as "Start"), the community assembly components of biomass change are displayed in the order of 1) species richness loss, 2) species identity loss, 3) species richness gain, 4) species identity gain, and 5) abundance change of persisting species (see Fig 2 for detailed explanations), reaching the predicted endpoint of absolute biomass highlighted with a black horizontal bar and labelled with the respective timespan (2 years, 7 years). Note that vectors for species identity and abundance change only expand along the y axis. Vectors are based on median predictions from linear mixed-effects models (see methods and Tables S20, S21). Vectors for high PSR (60 plant species) and low LUI (0.5) plots are colored in green, vectors for low PSR (monoculture) and high LUI (3.5) plots are colored in beige (see legend). See Fig S16 for an illustration of the underlying data spread after 7 years.

Response 2: For higher impact and therefore your own later benefit I would keep in mind that these equations are of course a tool utilized to place order on a biological process. Without a clear description of how to read this format of figure, you are selecting an audience who are already interested in Price equations rather than an audience who are interested in arthropod declines. The additional information in the caption is very welcome, but I would still add some very specific background to help us arthropod people out. Notably, which arrows are vectors, and what exactly are they showing? Many will be unfamiliar, and might not take the time to read up on Price equations and figures in order to understand this specific manuscript with a focus on arthropod declines.

Inflammatory Material:

None.

Some minor comments:

Throughout: Landscape 'management' can be for the benefit of biodiversity (eg active habitat restoration) as well as for activities that negatively impact biodiversity. For example, line 255: I hope that those 'protected grasslands' are not 'unmanaged', otherwise there is no purpose in the protected area.

#20. Response: Absolutely, such cases exist, we clarified that we're talking about intensive land use and not management per se. The referenced study, however, indeed reports declines from unmanaged, protected grasslands. They are simply high enough in the mountains to not require management (while outside of Central Europe, many grasslands do not require management, the referee is right that most Central European grasslands require management):

L93: Land-use intensification, including e.g. fertilization, frequent mowing and more intense grazing in grasslands, however, can homogenize arthropod communities^{27,31}, accelerating the global loss of species and ecosystem functioning^{13,14}.

L320: Accordingly, arthropod declines were previously not only reported from intensively managed or disturbed ecosystems, but also from protected and natural ecosystems such as unmanaged grasslands¹⁸ or tropical rainforests^{2,37}.

Reviewer 2: Annoying nitpicking really, but important for clarity: the referenced grassland are managed if they are protected, but they might not be posit

Line 45: Sure community homogenisation is itself the adaption to environmental change?

#21. Response: That is correct, but homogenization is an adaptation which reduces adaptability. Sure, one could argue that the communities showed the necessary adaptability to homogenize, but as homogenization progresses, the adaptability to future changes declines. See also (Gossner et al., 2023). We added the word "future" to our sentence on reduced adaptability (L45).

Reviewer 2: Agreed.

Line 64: 'Elusive changes' is vague and needs explanation.

#22. Response: We adapted the section accordingly, it now reads:

L65: However, even if there are no local declines in species richness, abundance or biomass, community (dis-)assembly must be taken into account to capture changes in species identities and dominance, potentially altering ecosystem functions provided by arthropods^{25,26}.

Reviewer 2: Makes sense.

Lines 64-70: Three concepts are very rapidly introduced here and require greater explanation each.

#23. Response: The sections has been expanded accordingly and now reads:

L68: This is for multiple reasons: i) Homogenization: Species richness measures may miss homogenizing effects on the functions present in the community, e.g. due to adaptation to specific land-use types of novel climatic regimes^{23,27}. ii) Trait shifts: Trait-based analyses suggest that anthropogenic global change may increase shares of small-bodied species in arthropod communities, possibly because species with smaller body sizes are better able to cope with diminishing, yet variable resource and habitat availability^{28,29}. iii) Abundance shifts: The ecological consequences of community turnover moreover depend on abundance, that is dominance changes in persisting species, and the abundance of lost and gained species^{26,30,31}. For example, formerly highly abundant species may not be lost entirely, but declining numbers could reduce their functional impact significantly²⁶. On the other hand, rare species with small contributions to the communities' functioning may be lost entirely, but the consequences for the net community functioning could be negligible^{25,26}. Indeed, rare species are generally at higher risk of declining than common or dominant species^{6,32} (but see refs. 10,21,33), potentially shifting the relationship between species richness and ecosystem functioning over time³⁴.

Reviewer: Glad to see this extended background. I would change 'dominance' to 'shifts in abundance ratios' or similar, because ecological dominance could have behavioral aspects over just abundance.

Line 78: 'dominance patterns of ecosystem functioning' – this doesn't make sense in terms of wording, but also there is probably too high an emphasis on ecosystem functioning given that this manuscript doesn't quantify it.

#24. Response: Agreed, this sentence needed revision. We also clarified further that arthropod biomass is a proxy for ecosystem functioning:

L82: In sum, combining the quantitative and qualitative perspective of abundance change and species identity turnover within community assembly may help elucidate shifts in community metrics, such as biomass, and therefore ecosystem functioning, that previously went unnoticed^{20,26,35,36}.

Reviewer 2: Again, I simply wouldn't have such emphasis on ecosystem functioning as it was not quantified. Your focus is biomass.

Lines 80-91: This section on plant species richness and land-use intensity reads a little glued-on to the prior introduction on arthropod declines. As this is a major focus of the manuscript, I would better integrate these concepts beginning at the start of the text.

#25. Response: We agree with the benefits of introducing the roles of plant species richness (resource and habitat) and land use intensity early on. Please find the introduction in the very first paragraph:

L55: While the causes of arthropod declines are often related to anthropogenic global change, including land-use intensification and subsequent loss of habitat and basal resource diversity^{2,4,12,13}, the consequences of their shrinking populations on ecosystem functioning are poorly understood¹⁴.

Reviewer 2: Glad to see this included early on.

Lines 112-113: A clear statement on hypotheses tested, and why, on herbivores and predators specifically is necessary.

#26. Response: Done, the section now reads:

L129: Assuming a more direct link of plant diversity to primary consumer diversity, such as herbivores, than to secondary consumer diversity, such as predators^{27,55}, we further added separate analyses for the community (dis-)assembly of herbivorous and predatory arthropods, using the Jena Experiment data.

Reviewer 2: Glad to see this but I would place the potential differences between herbivores and predators in the context of arthropod declines. There is evidence going either way as to whether predators and higher trophic level species are most vulnerable to perturbation, as they are reliant on all the species below them in trophic structure.

Line 133: Without going too much into Methods in the wrong place, there needs to be at least some indication of where these percentages are coming from.

#27. Response: Done, the section now reads:

L155: After seven years, over 90% of predicted local arthropod biomass loss was associated with species richness declines, while abundance losses of persisting species contributed only up to 8%.

Reviewer 2: A nice summary.

Lines 211-212: The 'worst' what? Needs clarification.

#28. Response: Agreed, the section now reads:

L260: If gains do not keep up with losses already now, further acceleration of species loss might be yet to come.

Reviewer 2: I think alluding to accelerated species loss is pretty tenuous, so I would simplify to "species losses are likely to continue" or similar.

Lines 215-220: Surely this highlights my previous thoughts about occasional arthropods visitors to tiny sampling plots? The better-sampled and larger plots of the Biodiversity Exploratories seem to have less of an impact of measured species richness?

#29. Response: As outlined in response #1.4, there is no doubt that sample coverage will generally influence the detection of rare species, including random visitors to the plots. We now acknowledge this issue with coverage analyses and an

extended discussion in our manuscript. This is, however, no problem in our study, since (i) all our analyses are based on constant sampling efforts within datasets, therefore relative changes of biomass and species richness remain valid, and (ii) the results reported here reflect on local community assembly, possibly the most important ecological scale, as it is the scale of most biotic interaction. Of course, if one were to increase the spatial scale indefinitely, the contribution of species turnover would diminish up to the point that only definite global extinctions would contribute to species loss and newly evolving species would contribute to species gain. The undoubtedly interesting question of scaling the ecological Price equation from local to regional to possibly continental scales lies outside of our scope here, but presents a logical next step for the research field. However, we would like to emphasize again that despite the more than 10-fold larger size of the Biodiversity Exploratories plots compared to the Jena Experiment, our results are consistent. Both relatively over time and in terms of contributions of species richness to turnover: 95% in the Jena Experiment, 93% in the Biodiversity Exploratories. This is a robust finding. We still agree that it is worth highlighting the questions of coverage and scale even further. Please refer to our extensive responses #1.2 and #1.4 and the following lines:

L204: Despite the differences in setup (experiment vs real world), spatial scale and location in the investigated research programs, our partitioning approach consistently showed that the vast majority of local arthropod biomass loss was linked to declines in species richness (Fig 4).

L380: As our results reflect local, community-scale processes, we encourage future research to investigate the scale-dependence of the ecological Price equation, potentially yielding different assembly mechanisms for e.g. whole regions or biomes³⁵.

Reviewer 2: I agree that constant sampling effort ensures that relative changes in biomass and richness are valid, and that the spatial scales could be considered appropriate to arthropod community assembly. I would mention the slight difference in insect taxa sampled at L204 also, which is beneficial to your argument, as if they do have slight different ecology and life history etc etc they are still contributing to the same general trend. At L380, it is an interesting point but again places this manuscript in the context of the uses of the Price equation and not specifically addressing the mechanisms of arthropod decline. For maximum impact in the debate on arthropod declines, I would try and keep the discussion relevant to the biological process and not the analytical method.

Lines 243-244: 'gained species even had above-average biomass, indicating their ecological dominance' this requires a reference if it is true.

#30. Response: Agreed, moreover the phrasing could use improvement:

L309: In plots of high land-use intensity, gained species even had above-average total biomass, possibly indicating the rise of abundant generalist species, establishing their ecological dominance²⁷ (Fig 3d).

Reviewer 2: My previous points on dominance are relevant here.

Lines 271-272: I don't think this comparison of differing point estimations is appropriate as they are separate models, there is no post-hoc test, and the confidence intervals are clearly overlapping.

#31. Response: The referee is right that the sentence required more careful and clear phrasing. Further, there was a mistake in the order of estimates in the graph, which we fixed now. The section now reads:

L339: In trend, plant diversity strengthened the role of species identity for biomass changes in both trophic guilds, but it only promoted rare species turnover significantly for predators (Fig 5c, d).

Reviewer 2: Agreed, makes more sense.

Line 279: There is no quantification of ecosystem functioning here, so it is impossible to prove links with ecosystem functioning.

#32. Response: The referee is right that there is no direct quantification of ecosystem functioning *sensu stricto* here. Similar as in response #24, we highlight that biomass is a proxy for functioning and phrase the sentence more carefully:

L347: Overall, we show that high plant diversity and low land-use intensity benefit multitrophic diversity and potentially biomass-mediated functioning^{27,39,62}.

Reviewer 2: Agreed, makes more sense.

Line 296: Trends in arthropod diversity/abundance/biomass are likely highly variable, so it doesn't make sense to compare potential trends between studies without specifying the type of habitat, scale and region.

#33. Response: While it is notable that trends show quite some consistency across regions and biomes, we specified the cited studies accordingly:

L369: Notably, previous long-term research across agricultural, grassland and forest sites suggests that arthropod declines started well before our study period^{2,12,19}, potentially explaining the already relatively minor role of rare species in early years of our study

Reviewer 2: Agreed. Declines probably started hundreds of years ago. It is good to be aware that this is still a very narrow snapshot.

Line 306: 'increasing plant diversity' needs clarification. Do you recommend active replanting of human-modified grasslands with additional plant species?

#34. Response: While it sometimes may be an option, indirect promotion of plant diversity in grasslands certainly is more realistic than active planting (in contrast to managed forests of course, but this is out of our scope). We split and modified the sentence to clarify:

L383: On a positive note, we clearly identify increasing plant diversity and decreasing land-use intensity as mitigating factors for arthropod declines and their community simplification. As plant diversity and decreasing land-use intensity of grasslands are closely intertwined⁶¹, land-use extensification is a promising avenue for fostering arthropod-mediated ecosystem functioning and resilience in the face of future environmental change.

Reviewer 2: It's good to see a quick recommendation, but it would probably make sense to justify whether intensification is broadly considered better for species/ecosystem conservation than intensification. Both have their negatives of course.

Referee #2 (Remarks on code availability):

The code is accessible, moderately commented, and I was able to follow. It was easy to reproduce the core analysis. The input data has been pre-processed into pairwise comparisons between sites. I would argue that to be fully transparent, the raw arthropod species-level data should be available plus the code for manipulating that raw data.

#35. Response: We are happy to hear that the code is reproducible. We previously provided a subsample of the raw data because the Price equation processing needs extensive time, but we are happy to provide the full dataset now. The respective code was already a part of the R code provided in the github repository, but all code is extended and updated now, reflecting the revised statistical analyses.

Reviewer #2 (Remarks on code availability):

The code is available on GitHub and I was able to follow the analysis. The updated code is lesser commented than the previous version but is clearly laid out. A README is included.

Reviewer #3 (Remarks to the Author):

I reviewed an earlier version of this manuscript showing that overall declines of arthropod biomass in grasslands in two European decade-long studies were related primarily to reductions in species richness. The authors have conducted a comprehensive set of reanalyses of their data in response to the previous round of review, and have made extensive changes to their text, figures and supporting material.

The reviewers had all found interest and value in the previous submission, but had raised significant points especially about the analyses and interpretations. The authors have taken these criticisms seriously and it is reassuring that based on the reanalyses and checks the results and conclusions remain consistent. Perhaps the main improvement is that the approaches and details of the methods and assumptions are now presented much more transparently and completely, increasing the ability of the reader to assess these.

I do not have additional substantial comments on the work beyond my previous assessment. The importance of the losses to species richness and the approach used remain interesting, and the authors have addressed my main concerns. My assessment from the detailed responses to the other reviewers is also that genuine efforts have been made to address all of these, and that the results remain consistent.

I appreciate the additional material that has been added to the supplementary material. I realized when reading the responses to referees that in the main text before the methods there is no mention of the taxonomic groups that were included (they are just referred to as "arthropods"). These are not mentioned until L430 ("the highly diverse order of Coleoptera, the herbivore-dominated Hemiptera (excluding Sternorrhyncha), and the predatory Araneae" and then Hymenoptera in Jena and Orthoptera in the Biodiversity Exploratories). I think that if possible it would be useful to mention the taxa somewhere in the main text, before the Methods, to give an idea of the guilds or functional groups included.

I agree with referee 2 that Fig 4 requires very careful attention to interpret. The new version includes arrows on the vectors of the individual effects. I infer from this that at least in the case of species identity loss (the bright green lines) this is leading to increases in biomass because rare species with low overall biomass have been lost – after both 2 years and after 7 years?

Minor points / typos

Line 70: I think this should be "or novel climatic regimes" (not "of")

Line 228: "arthropod species in grasslands were previously shown to decline the fastest". Is there a word missing? "rare arthropod species"??

Line 229: "shown to decline", "their diminishing contribution" (or "their diminished contribution")

Reviewer #1 (Remarks to the Author):

I reviewed an earlier draft of this manuscript, and then concluded that the paper provided valuable insights and was worthy of eventual publication, although I had some specific qualms that I wished to see addressed (as did the other reviewers). Most of those points are substantially addressed in the current manuscript. The authors do a much better job of making it clear that the changes being described are inferences from community samples, and that as such claims about the dynamics of the actual communities themselves -- let alone about regional or global arthropod dynamics, need to be expressed with appropriate caveats. The paper as it stands is generally clear, well argued and well supported by two excellent datasets. It makes a strong contribution to an ongoing debate.

#1 Response: We thank the referee for acknowledging the improvements in our revised manuscript and its overall significance. The previously provided review helped a lot in identifying and clarifying possible caveats.

My only somewhat-major concern remaining at this point is the assertion (e.g. lines 234-5) that constant sampling effort, the restriction of sampling to "dry and windless" days (line 415) and the use of the "control" of examining turnover within-site between replicate samples solves the problem of pseudo-extinctions and immigrations. Constant effort sampling is commendable, but it doesn't equate to constant intensity sampling: subtle differences in weather or timing can translate to large shifts in insect catches. Restricting samples to "dry and windless" days (line 415) helps somewhat, but differences in temperature, humidity, and random events such as emergence or migration events can produce substantial stochasticity in insect activity and thus catch sizes, despite constant effort. Their reply to my earlier query about incorporating weather data into their analyses ("...our analysis is not aiming at the external drivers of arthropod declines") misses the point in this respect: CLIMATE may be a driver of insect decline, but WEATHER on the day of sampling is a sampling issue.

#2 Response: We thank the referee for clarifying that the raised concerns refer to the weather and other factors (such as random events) that create stochasticity, and not to the climate. To address this, we now clearly report our standards for suitable sampling weather and provide detailed tables of weather properties during the sampling day (i.e. air temperature and humidity) in the supplementary material. As reflected in the annual values and the low standard deviation (~2°C) among years, we sampled under nearly standardized weather conditions. Further, we now put more emphasis on the fact that we use moving average analyses, which were chosen exactly for the point of buffering stochasticity. As outlined in the manuscript, the moving average analyses reduce the impact of single sampling years and identify replicated, robust temporal patterns. Notably, any remaining stochasticity would rather be a concern if we did not find consistent temporal patterns among analyses, indicating potentially blurred relationships – but despite all natural variability among samples, we found clear patterns. Please find the revised sections on sampling weather and the robustness of the analyses in the following lines:

L120: Since high inter-annual variability of arthropod diversity, ecosystem functioning and environmental conditions was previously reported in both research programs^{43,50}, we modelled linear temporal trends of each component based on pairwise comparisons using a restricted moving average approach. For this, we pooled all available pairs for each time span (moving average) that include any of the first five years as baseline (restriction). We thus generated more generalizable results, reducing the sensitivity to the first sampling year and single years in general (with e.g. climatic extremes or random events affecting sample size) while also reflecting systematic temporal trends⁵¹.

L163: Yet, reducing the dependence on the first sampling year (baseline) and smoothing, but not removing, extreme years with e.g. climatic anomalies, makes detected trends more generalizable and robust⁵¹ (see Extended Data Fig. 4, 5, Supplementary Notes 1, 2, Fig. S1, S2 for further sensitivity analyses).

L373: All samplings were conducted under standardized conditions (dry and windless, after morning dew had dried; see Tables S25, S26 for details on mean temperature and humidity on the sampling day).

I am also puzzled by how the restriction of analyses to species occurring in at least 10% or 30% of plots helps address this problem. The finding (lines 271-272) that ignoring rare species reduces the proportional effect of species richness change on biomass seems to confirm the reviewers' concern that much of the turnover may down to poorly sampled species (although, to be fair, rare species are also mostly likely to have genuine local extinctions). Meanwhile, the fact that this decline in relative richness effects is associated with increasing relative contributions of species identity and abundance (lines 272-273) is surely unremarkable (as these are relative contributions). Nor is it surprising that removing the rarest detected species from consideration decreased the measured contribution or rare species turnover (277-278).

#3 Response: We agree that this sensitivity analysis does not directly address the issue of weather conditions during sampling. Our response during the first revision was misleading in this regard (for the problem of possible impacts of weather and how we addressed this issue, please refer to response #2). Removing rare species in terms of absolute occurrences (occurring in less than 10% or 30% of the plots), rather addresses the issue of: how robust or variable are results when recalculating average biomass values (species richness) and deviations from it (species identity) after reducing the “noise” in species turnover, i.e. removing species which may have “randomly” occurred in our study plots (but note that 30% of plots is a conservative cutoff, “random” might not apply to species occurring in e.g. 25% of plots). The findings, however, are not a self-evident fact, since all Price components are calculated newly and always refer to the (newly calculated) community mean. Indeed, there was a small mistake in the previous version of the manuscript: in the Exploratories, both relative and absolute contributions of species identity to biomass change decreased (moderately) when removing rare species – contrasting to the moderate increase of relative and absolute contributions of species identity in the Jena Experiment. Therefore, we argue that it is reassuring and transparent in terms of the study caveats that artificially reducing “noise” does only moderately change the relative contributions of the Price components and that the observed

moderate changes in the importance of abundance are similar to previously observed effects when increasing the spatial scale (see e.g. Hillebrand et al. 2018), but species identity effects depend on the study design. Please find the clarified section in the following lines:

L253: Moreover, supplementary analyses show that removing species occurring in less than 10% or 30% of plots reduced biomass change associated with species richness turnover, but the relative contribution of abundance change increased in both research programs by 1 - 2%, and species identity effects increased by ~1% in the small plots of the Jena Experiment (Supplementary Note 4, Fig. S6, S7, Tables S20, S21). Removing the rarest detected species can thus be interpreted as similar to increasing the sample size: while stochastic species turnover decreases, shifts in abundance and species identity gain in importance³⁰. Yet, only in the Biodiversity Exploratories, removing the rarest detected species decreased the contributions of rare species turnover. This emphasizes differences in the study designs, with the Biodiversity Exploratories spread over a large geographic range, and the Jena Experiment covering one field site; while 72% of species occurred in at least 30% of the plots in the Jena Experiment, only 57% of species did so in the Biodiversity Exploratories. Accordingly, excluding species that occur in few plots was more impactful in the Biodiversity Exploratories.

Some minor points:

Lines 68-69: LOCAL species richness misses homogenization effects (but these are reflected in coarser-scale richness).

#4 Response: Agreed and inserted.

87-89: Awkward sentence; re-phrase.

#5 Response: Done. The sentence now reads:

L86: Widespread negative effects of climate change on biodiversity are well documented^{2,3,11,37}, but local plant diversity declines and land-use practices also affect ecosystems^{1,13,38}.

109: Awkward phrase; "into the underlying community (dis-)assembly PROCESSES"? (or components?)

#6 Response: Absolutely, there was a word missing. We added "processes" as suggested.

179-182: I'm not convinced that 8 years (2002-2010) is sufficient time to pay off the extinction debts of a 1 ha experimental site. The cited reference shows community change slowing somewhat, but it would likely continue for some time (although I'd expect longer in terms of colonisation credits than for extinction debts). Also: previously (line 180).

#7 Response: Yes, although the study supports our conclusions, it is likely that the system has not fully transitioned by that time. We phrased the sentence more carefully and corrected the typo:

L172: Extinction debts due to the establishment of the Experiment on a previously agricultural site in 2002 are also unlikely to have substantially influenced our results, as they were probably largely paid off by 2010, when grassland arthropod communities were established⁵⁷.

Figure 4 (and Line 209): This figure is nicely information-dense, and useful, although I didn't entirely understand it. For instance, what does statement "vectors for species identity and abundance change only expand along the Y axis" (lines 331-332) mean? Also, with two 11-year time series, it's unclear why the focus here should only be on 7 years of change. Presumably this is linked to the limits of the Jena moving average comparison (lines 484-5; but if so, this should be explicitly stated). Would it be possible to assess the full 11-year temporal dynamics using the fixed baseline time series (in supplemental materials)?

#8 Response: We thank the referee for highlighting points of confusion in this graph. Since all three referees raised concerns about the clarity of the graph and the caption, we added a conceptual vector graph including illustrations of the observed community change (similar as in Fig. 1) in Extended Data Figure 1 and adapted the caption in the main (see below). Further, we added a graph as suggested by referee 1 to the supplement, covering the total timespan (fixed baseline; Figure S11).

L601: FIGURE 4 Arthropod biomass declines are primarily associated with species richness loss, while losses and gains of species with non-average total biomass (species identity) contribute disproportionately to arthropod biomass change under high plant species richness (PSR; Jena Experiment) and low land-use intensity (LUI; Biodiversity Exploratories). Modelled change of arthropod biomass (y axis) and species richness (x axis) per replicate per plot after 2 and 7 years (restricted moving average) in dependence of PSR and LUI. Starting at the average community biomass and species richness value in the first year (Baseline), the community assembly components of biomass change are displayed as vectors (arrows) in the order of 1) species richness loss, 2) species identity loss, 3) species richness gain, 4) species identity gain, and 5) abundance change of persisting species (see Fig. 2 for detailed explanations), reaching the predicted comparison community values of absolute biomass and species richness after the respective time spans of 2 and 7 years. Most arthropod biomass change is associated with species richness change, but the vertical vectors of species identity and abundance change show that species richness change alone does not explain all biomass change. Vectors are based on median predictions from 1,000 linear mixed-effects models (see methods and Tables S22, S23). Vectors for high PSR (60 plant species) and low LUI (0.5) plots are colored in green/blue, vectors for low PSR (monoculture) and high LUI (3.5) plots are colored in beige/brown (see legend). See Extended Data Fig. 10 for an illustration of the underlying data spread after 7 years. 7 years is the maximum replicated moving window in the Jena Experiment, see Supplementary Note 1 for information on the maximum time span of 10 years (Fixed baseline).

225-226: "rare species" are usually determined in terms of abundance. "below average total biomass" and "above average individual biomass" gets close to that, as you could back-calculate abundance by dividing total biomass by individual biomass. But presumably you have access to the actual abundance data, which would define species rarity much more precisely. Why not look at that?

#9 Response: Using the partitioning of biomass change, the manuscript focuses on relative biomass contributions to the community – and rarity is thus defined via biomass contributions. However, we agree that it is reassuring to check whether abundance patterns match our conclusions drawn from biomass distributions. We added species rank abundance curves to the supplementary material (Fig. S3). Find the added relevant method section in the following lines:

L200: Yet, while species loss and associated biomass loss increased in later years, the absolute offsets by lost rare species stagnated or declined, reducing relative offsets to 0.4 (-0.1, 2.8) - 3.4% (0.4, 9) and gained species had almost entirely average biomass (see Fig. S3 for additional rank abundance curves).

229: diminishING?

#10 Response: Yes, we thank the referee for catching that. It is corrected.

304: Is "assure" the correct word? "be certain" or "assume" perhaps?

#11 Response: "assume" is a better choice, yes, we adapted the sentence accordingly.

362: Is it really true that "communities ... homogenized almost completely in later years"? If so, it would be good to see that quantified in some manner (e.g. through between-sample community similarity or difference metrics)

#12 Response: We thank the referee for pointing out this unclear sentence. It is indeed about biomass distributions, not any other measures of homogenization. We adapted the section, it now reads:

L315: However, communities shrank and their biomass distribution homogenized almost completely in later years, with biomass declines primarily associated with species richness declines—a pattern which was consistent among the different setups and scales of the Jena Experiment (experiment, small scale) and the Biodiversity Exploratories (real world, larger scale), and which likely transfers to many shrinking biotic communities³⁰.

468-469: Unclear what the meaning of "fallback to the 'Chao' estimator" means in this context. Fall back under what circumstances?

#13 Response: In cases of single species making up more than 50% of the sample, Chao seems to be the more robust estimator. Considering the comments from referee 2, we suggest avoiding unnecessary detail here, and rephrase the sentence to a more concise and entirely reproducible report of methods used (the “Best” estimator in the coverage estimation of entropart handles all cases fully automated):

L427: To assess possible changes in sampling coverage over time, we estimated the sample coverage of each sample (replicate) using the “Coverage” function with the “Best” estimator in the entropart package⁷⁷.

Reviewer #2 (Remarks to the Author):

Thank you for sending me an update on this manuscript. I am pleased to read that the Authors of this manuscript have made substantial progress. I am the original Referee 2 and as such I write this review mostly as if this manuscript were a resubmission to the same journal. I am glad that my comments have been helpful so far.

I do believe that the Editors have made the correct decision in transferring the manuscript to Nature Ecology & Evolution, which is of course another top-level publication. The subject matter, global arthropod declines, is of utmost importance and should be addressed in such top publications. This manuscript describes important and novel aspects of biodiversity change, from a good dataset, and thus should be published when finalized. The resampling remedy employed by the Authors does seem to have technically addressed the issue of pseudoreplication at the data modelling stage. I still believe that to keep reporting back-and-forth between the Jena and Biodiversity Exploratories datasets is a little limiting on the Reader’s comprehension of an overall bigger picture, and that some of the analytical methods (which would probably show the same trends, only in a different format) would be a little easier to comprehend through Generalized Additive Mixed Modelling. This is a complex manuscript, with a lot of concepts and some concepts (eg Price equations) which are probably new to many Readers interested in arthropod declines and thus really need to be explained 1) in sufficient depth and 2) in constant relation to the actual subject focus of arthropod ecology and threats. Those are, however, mostly my subjective opinions on structure rather than a criticism of integrity. I have commented on the Authors’ responses below. I have reduced my original comments to save space and maintain dialogue.

#14 Response: We thank the reviewer once again for the helpful review, we appreciated the efforts to really help developing the manuscript. We are happy to read that most concerns have been addressed and we adapted our manuscript once more considering the newly raised points. We only kept such points from the second round of review in our response, which identified new or persisting problems. Considering the comments above, find adapted sections on the Price equation and implemented modelling methods and their relevance for arthropod declines in the following lines:

L107: The ecological adaptation partitions changes in ecosystem functions^{25,26,47}, or in our case biomass^{48,49}, between two communities into the underlying community (dis-)assembly processes, separating effects of average species turnover (species richness) from non-average species turnover (species identity) and effects independent from species turnover (here: Abundance change). Specifically, the five components are: i+ii) species losses and gains assuming that all species undergoing turnover have average biomass relative to their respective communities (expected effect of species richness), iii+iv) the difference between the expected and observed biomass change associated with species turnover, i.e. the deviation of lost and gained species from the average biomass of their respective communities (species identity of lost and gained species), and v) changes in abundance of persisting species²⁵ (Fig. 1, Extended Data Fig. 1, 2; see Supplementary Methods for the mathematical equation). In declining arthropod communities, the ecological Price equation may thus help identifying whether species loss per se or more subtle changes in community composition underpin biomass loss. Since high inter-annual variability of arthropod diversity, ecosystem functioning and environmental conditions was previously reported in both research programs^{43,50}, we modelled linear temporal trends of each component based on pairwise comparisons using a restricted moving average approach. For this, we pooled all available pairs for each time span (moving average) that include any of the first five years as baseline (restriction). We thus generated more generalizable results, reducing the sensitivity to the first sampling year and single years in general (with e.g. climatic extremes or random events affecting sample size) while also reflecting systematic temporal trends⁵¹.

Reviewer 2: Ok sure, it does seem irrelevant to group plots, I understand that decision. I would though suggest that getting into the mathematical quantification of coverage/entropy may be a bit of a rabbit hole – most estimators assume a closed system, and my previous points largely pertained to highly dispersive species (for example, adult beetles flying) around and unpredictably landing in small sampling sites. Therefore, I don't think those coverage values of ~90% are very meaningful, as each small site is an open system which will experience many visitors. My advice would be to place emphasis on biological knowledge and admission of caveats on arthropod sampling given the sampling design and such arthropod behavior. I think arthropod scientists would generally accept that, and it would probably save you undue hassle in attempting to mathematically define coverage! The mobility part requires a brief description of what A-D and 0-1 mean as dispersal measures, as not all Readers will access those associated references.

#15 Response: Agreed, careful interpretation is key. We suggest to keep the coverage analysis as it is discussed only relatively, i.e. we get an idea that it is decreasing over time and the other referees supported the revisions. However, the referee is right to not emphasize these values too much. We added further sections on the mobility issue in the following lines:

L214: Yet, we emphasize that first, since our sampling effort was constant over time, declining detections still reflect shrinking abundances and biomass, indicating that most species are becoming rarer in absolute terms; and second, declines in the turnover of rare species were particularly strong in the Biodiversity Exploratories, where sampling coverage was higher than in the Jena Experiment and only decreased by one percent (Fig. 3c, d; S4). Further

supplementary analyses showed that rarely detected species were not just highly mobile “tourists” in our plots, but rather the opposite: commonly detected species tended to show higher mobility (Fig. S5, Tables S18, S19). This implies that in the open systems of our plots, turnover of common species may be amplified by mobile, visiting species to some degree (see Supplementary Note 4, Fig. S6, S7 for additional sensitivity analyses on the robustness of observed patterns when reducing the analysis to species that occurred in at least 10% or 30% of all plots per year per research program).

Reviewer 2: I think it important to note that occasional climatic occurrences do have biological meaning also beyond statistical noise, for example the the recent and already referenced Sharp et al. 2025 paper on El Nino (admittedly from the tropics, and different threats). It is undoubtedly difficult to separate the noise from the meaning. That doesn't invalidate your justified methods. I believe it just requires a sentence or two of written recognition.

#16 Response: Agreed, it is an important strength of the restricted moving average that it reduces the impact of extreme years but does not remove it. The adapted section now reads:

L163: Yet, reducing the dependence on the first sampling year (baseline) and smoothing, but not removing, extreme years with e.g. climatic anomalies, makes detected trends more generalizable and robust⁵¹ (see Extended Data Fig. 4, 5, Supplementary Notes 1, 2, Fig. S1, S2 for further sensitivity analyses).

Reviewer 2: I agree with Reviewer 3 in that the differences in scale required addressing, but that relates to my points that without some direct methodological integration of the two datasets, the Authors and Reader must always consider one and then the other. I don't believe that the two actually should be integrated, given the inherent differences in sampling, scale and coverage. I mean that that is a 'natural' limitation in this analytical design which probably, to some degree to be decided by the Editors, reduces the relative impact. What I mean is evident in the above text: “the different setups and scales of the Jena Experiment (experiment, small scale) and the Biodiversity Exploratories (real world, larger scale)”. I do not disagree that species richness losses largely underpin biomass reduction. Your results certainly support that conclusion, and that conclusion is also in line with much broader understanding that arthropod species richness declines accompany vegetation homogenization. The clarifications that the results pertain to Central European grasslands improve the accuracy of the text.

#17 Response: Agreed.

Reviewer 2: I do agree with your points but going back to arthropod biology as well as experimental design, we must also recognize that responses can differ at the family and even order levels due to differences in physiology and life histories. Inclusion of the taxon summaries is certainly helpful for Readers to make their own informed decisions on the impact. Again, I think those interested in arthropod biodiversity declines are broadly accepting

of these difficulties and they don't detract from your analyses, just need a little recognition in the text.

#18 Response: Agreed, the differing taxonomic coverage is now referenced in the main text:

L98: One time series (2010 – 2020; Coleoptera, Hemiptera, Araneae, Hymenoptera) was collected from the Jena Experiment, an experimental grassland site in Central Germany, comprising 80 small-scale plots (5 x 6 m) along a controlled gradient of plant species richness⁴⁴. The other time series (2008 – 2018; Coleoptera, Hemiptera, Araneae, Orthoptera) comes from 150 grassland plots of larger size (50 x 50 m) in the Biodiversity Exploratories, a network of real-world farmed grasslands spanning a wide range of management practices in three geographic regions across Germany⁴⁵.

L183: Despite the differences in setup (experiment vs real world), spatial scale, location and taxonomic coverage in the investigated research programs, our partitioning approach, that is the ecological Price equation, consistently showed that the vast majority of local arthropod biomass loss was linked to declines in species richness (Fig. 4).

Reviewer 2: Glad to see this information included. As another Reviewer also stated, the concept of Price equations will be new to many interested in arthropod declines and so they must be generally understandable without reference outside of this manuscript. There still isn't really any information given on LUI. Why not just a single sentence for understanding outside of a modelling context, for example "value X = no fertilization, grazing or mowing and value y = grazing or mowing to a barren state.." ?

#19 Response: Good idea, the provided information on the LUI now reads:

L360: Land-use intensity (LUI) values were calculated based on the combined intensities of fertilization, grazing and mowing, with LUI values ranging between 0.5 (e.g. less than one mowing event per year and no fertilization or low livestock densities) and ~3.5 (e.g. frequent mowing and high fertilization or intense grazing)⁵⁴.

Reviewer 2: I believe the first point has been addressed. The Authors have also provided background information on their 10% and 30% thresholds, more as sensitivity analysis to noise rather than specifically ecosystem-rare species. This makes more sense in the context of their analysis. Given that, and seconding an earlier comment of mine, I wouldn't put too much emphasis on the mathematical quantification of coverage in this open system ('open' because of how small some of the sites are). Coverage could instead be discussed and justified in relation to the sampling methods and arthropod behaviors.

#20 Response: As pointed out in response #15, we agree to not put too much emphasis on the mathematical quantification of coverage and rather discuss arthropod mobility and detection probabilities.

Reviewer 2: I understand that, but this is another case where the decisions made are based on Price equations, rather than arthropod ecology and associated background knowledge. For example, in the context of the title where 'identity' is used without reference to Price equations, the word to me implies taxonomic identity.,

#21 Response: Agreed, we believe that the updated title and introduction section (see response #14) remedy this issue.

Response 2: For higher impact and therefore your own later benefit I would keep in mind that these equations are of course a tool utilized to place order on a biological process. Without a clear description of how to read this format of figure, you are selecting an audience who are already interested in Price equations rather than an audience who are interested in arthropod declines. The additional information in the caption is very welcome, but I would still add some very specific background to help us arthropod people out. Notably, which arrows are vectors, and what exactly are they showing? Many will be unfamiliar, and might not take the time to read up on Price equations and figures in order to understand this specific manuscript with a focus on arthropod declines.

#22 Response: We thank the referee for highlighting persisting points of confusion in this graph. We agree that more help for the reader was needed to grasp the information, even when unfamiliar with the Price equation. Since all three referees raised concerns about the clarity of the graph and the caption, we added a conceptual vector graph including illustrations of the observed community change (similar as in Fig. 1) in Extended Data Figure 1 and adapted the caption in the main (see response #8 for further details).

Reviewer 2: Annoying nitpicking really, but important for clarity: the referenced grassland are managed if they are protected, but they might not be posit

#23 Response: We suspect that some part of the comment is missing here, but we believe we agree and suggest to adapt the sentence avoiding any use of "unmanaged":

L293: Accordingly, arthropod declines were previously not only reported from intensively managed or disturbed ecosystems, but also from protected and natural grasslands¹⁸ or tropical rainforests^{2,37}.

Reviewer: Glad to see this extended background. I would change 'dominance' to 'shifts in abundance ratios' or similar, because ecological dominance could have behavioral aspects over just abundance.

#24 Response: The word "dominance" is advisable in this context because it is used in our central reference for biodiversity turnover and the contributions of species richness, abundance and species identity (Hillebrand et al. 2018). We adapted the sentence to be more clear:

L73: Abundance shifts: The ecological consequences of community turnover moreover depend on abundance changes in persisting species, i.e. shifts in dominance³⁰, and the abundance of lost and gained species^{26,31,32}.

Reviewer 2: Again, I simply wouldn't have such emphasis on ecosystem functioning as it was not quantified. Your focus is biomass.

#25 Response: True, we phrased the sentence more carefully, avoiding deletion of the subject of ecosystem functioning, as it is a key reason why it matters what we present in this manuscript:

L81: In sum, combining the quantitative and qualitative perspective of abundance change and species identity turnover within community assembly may help elucidate shifts in community metrics, such as biomass, and potentially associated ecosystem functioning, that previously went unnoticed^{20,26,30,36}.

Reviewer 2: Glad to see this but I would place the potential differences between herbivores and predators in the context of arthropod declines. There is evidence going either way as to whether predators and higher trophic level species are most vulnerable to perturbation, as they are reliant on all the species below them in trophic structure.

#26 Response: We thank the referee for this valid hint, we adapted the section accordingly:

L132: Assuming a more pronounced decline of secondary consumers, such as predators, due to bottom-up effects^{7,21,43}, but a more direct link of plant diversity to primary consumers, such as herbivores^{27,55}, we further added separate analyses for the community (dis-)assembly of herbivorous and predatory arthropods, using the Jena Experiment data.

L260: If gains do not keep up with losses already now, further acceleration of species loss might be yet to come.

Reviewer 2: I think alluding to accelerated species loss is pretty tenuous, so I would simplify to "species losses are likely to continue" or similar.

#27 Response: Agreed, done.

Reviewer 2: I agree that constant sampling effort ensures that relative changes in biomass and richness are valid, and that the spatial scales could be considered appropriate to arthropod community assembly. I would mention the slight difference in insect taxa sampled at L204 also, which is beneficial to your argument, as if they do have slight different ecology and life history etc etc they are still contributing to the same general trend. At L380, it is an interesting point but again places this manuscript in the context of the uses of the Price equation and not specifically addressing the mechanisms of arthropod decline. For maximum impact in the

debate on arthropod declines, I would try and keep the discussion relevant to the biological process and not the analytical method.

#28 Response: Yes, as outlined in response #18 we now mention the differing taxonomic coverages at the suggested point. Regarding the comment on the Price equation, we adapted the sentence to reflect the relevance for the arthropod change:

L335: As our results reflect local, community-scale processes, we encourage future research to apply the ecological Price equation to temporal arthropod biomass and diversity change at larger spatial scales, potentially yielding different assembly mechanisms for e.g. whole regions or biomes³⁰.

L309: In plots of high land-use intensity, gained species even had above-average total biomass, possibly indicating the rise of abundant generalist species, establishing their ecological dominance²⁷ (Fig 3d).

Reviewer 2: My previous points on dominance are relevant here.

#29 Response: Yes, in this case “ecological dominance” is maybe interpreting too much and should be replaced with “numerical dominance” (see response #24).

L383: On a positive note, we clearly identify increasing plant diversity and decreasing land-use intensity as mitigating factors for arthropod declines and their community simplification. As plant diversity and decreasing land-use intensity of grasslands are closely intertwined⁶¹, land-use extensification is a promising avenue for fostering arthropod-mediated ecosystem functioning and resilience in the face of future environmental change.

Reviewer 2: It's good to see a quick recommendation, but it would probably make sense to justify whether intensification is broadly considered better for species/ecosystem conservation than intensification. Both have their negatives of course.

#30 Response: We are not entirely sure whether we understand the comment, we assume that the referee means that also extensification can have negative impacts on biodiversity/ecosystem conservation. We agree that some level of land use is often required for biodiversity protection and clarified accordingly:

L340: As plant diversity and decreasing land-use intensity of grasslands are closely intertwined⁶¹, land-use extensification—but not abandonment⁶⁸—is a promising avenue for fostering arthropod-mediated ecosystem functioning and resilience in the face of future environmental change.

Reviewer #3 (Remarks to the Author):

I reviewed an earlier version of this manuscript showing that overall declines of arthropod biomass in grasslands in two European decade-long studies were related primarily to reductions in species richness. The authors have conducted a comprehensive set of reanalyses of their data in response to the previous round of review, and have made extensive changes to their text, figures and supporting material.

The reviewers had all found interest and value in the previous submission, but had raised significant points especially about the analyses and interpretations. The authors have taken these criticisms seriously and it is reassuring that based on the reanalyses and checks the results and conclusions remain consistent. Perhaps the main improvement is that the approaches and details of the methods and assumptions are now presented much more transparently and completely, increasing the ability of the reader to assess these.

I do not have additional substantial comments on the work beyond my previous assessment. The importance of the losses to species richness and the approach used remain interesting, and the authors have addressed my main concerns. My assessment from the detailed responses to the other reviewers is also that genuine efforts have been made to address all of these, and that the results remain consistent.

#31 Response: We thank the referee for acknowledging our efforts and for contributing to improving the manuscript.

I appreciate the additional material that has been added to the supplementary material. I realized when reading the responses to referees that in the main text before the methods there is no mention of the taxonomic groups that were included (they are just referred to as “arthropods”). These are not mentioned until L430 (“the highly diverse order of Coleoptera, the herbivore-dominated Hemiptera (excluding Sternorrhyncha), and the predatory Araneae” and then Hymenoptera in Jena and Orthoptera in the Biodiversity Exploratories). I think that if possible it would be useful to mention the taxa somewhere in the main text, before the Methods, to give an idea of the guilds or functional groups included.

#32 Response: Agreed, that is a good idea. We now provide this information in the introduction and further highlight that taxon coverages differed between the time series following the recommendation by referee 2 (see response #18).

I agree with referee 2 that Fig 4 requires very careful attention to interpret. The new version includes arrows on the vectors of the individual effects. I infer from this that at least in the case of species identity loss (the bright green lines) this is leading to increases in biomass because rare species with low overall biomass have been lost – after both 2 years and after 7 years?

#33 Response: We thank the referee for highlighting persisting points of confusion in this graph. Yes, the interpretations of the species identity vectors are correct. However, since all three referees raised concerns about the clarity of the graph and the caption, we added a conceptual vector graph including illustrations of the observed community change (similar as in Fig. 1) in

Extended Data Figure 1 and adapted the caption in the main (see response #8 for further details).

Minor points / typos

Line 70: I think this should be “or novel climatic regimes” (not “of”)

#34 Response: Yes, done.

Line 228: “arthropod species in grasslands were previously shown to decline the fastest”. Is there a word missing? “rare arthropod species”??

#35 Response: Yes, good catch, done.

Line 229: “shown to decline”, “their diminishing contribution” (or “their diminished contribution”)

#36 Response: Yes, done.

References

Hillebrand, H., Blasius, B., Borer, E. T., Chase, J. M., Downing, J. A., Eriksson, B. K., Filstrup, C. T., Harpole, W. S., Hodapp, D., Larsen, S., Lewandowska, A. M., Seabloom, E. W., Van De Waal, D. B., & Ryabov, A. B. (2018). Biodiversity change is uncoupled from species richness trends: Consequences for conservation and monitoring. *Journal of Applied Ecology*, 55(1), 169–184. <https://doi.org/10.1111/1365-2664.12959>